# Translation in *Bacillus subtilis* is spatially and temporally coordinated during sporulation

Olga Iwańska [1,4], Przemysław Latoch [1,4], Natalia Kopik[2], Mariia Kovalenko [1], Małgorzata Lichocka[1], Remigiusz Serwa [3] & Agata L. Starosta [1] ✉

The transcriptional control of sporulation in *Bacillus subtilis* is reasonably well understood, but its translational control is underexplored. Here, we use RNA-seq, ribosome profiling and fluorescence microscopy to study the translational dynamics of *B. subtilis* sporulation. We identify two events of translation silencing and describe spatiotemporal changes in subcellular localization of ribosomes during sporulation. We investigate the potential regulatory role of ribosomes during sporulation using a strain lacking zinc-independent paralogs of three zinc-dependent ribosomal proteins (L31, L33 and S14). The mutant strain exhibits delayed sporulation, reduced germination efficiency, dysregulated translation of metabolic and sporulation-related genes, and disruptions in translation silencing, particularly in late sporulation.

*Bacillus subtilis* is by far the best understood spore forming bacterium. Spores are mostly metabolically inactive, extremely resistant cells that are formed in response to nutritional stress. In the process of sporulation, *B. subtilis* divides asymmetrically to form two cells with differing morphology and cell fate – a small forespore and a larger mother cell. This process is tightly controlled and highly trackable, thanks to which it has been used as an excellent model to study cell development and differentiation, intracellular signalling or gene expression regulation for the past few decades[1–3].

Transcriptional control of sporulation has been described extensively and is reasonably well understood[4–7]. The decision to sporulate is not one that is made lightly by the bacterial population as this process requires fundamental changes in the metabolism and morphology that take approximately 7 h at 37 °C[8]. In fact, *B. subtilis* displays cannibalistic behaviour with the secretion of two sporulation killing factors, Skf and Sdp, responsible for killing nonsporulating cells. This ensures nutrient release for the remaining cells expressing resistance genes, which delays the decision to sporulate and allows the cells to resume normal growth should the environmental conditions improve[9]. However, once the cells decide to sporulate, the transition from vegetative growth to sporulation is under the control of Spo0A transcriptional master regulator. High levels of phosphorylated Spo0A activate transcription of key sporulation factors, *spoIIA*, *spoIIE* and *spoIIG*, which in turn regulate the expression of compartment specific RNA polymerase

sigma factors – σF and σE – responsible for further activation of sporulation genes in the forespore and mother cell respectively[3,10]. Polar septum formation is also a signal to recruit SpoIIIE protein which actively transports bacterial chromosome into the forespore[11]. After asymmetric division and chromosome translocation, genes in σF and σE regulons drive metabolic and morphological changes to the cell which conclude in forespore engulfment, creating a cell-within-a-cell state. This is required for activation of the late sporulation spore specific sigma factor σG responsible for spore maturation, spore DNA protection via Ssp proteins and preparation for germination[12]. Finally, the late mother cell σK activation, regulated by σG, results in spore coat and cortex maturation, preparation for germination and eventually, mother cell lysis[3,8].

Translation regulation in *B. subtilis* has gained more attention in the recent years with a number of publications describing translation during germination[13,14], during or at the entry into quiescence[15,16] or transcription-translation uncoupling[17]. Translational control during sporulation remains an understudied area, although the role of post-transcriptional regulation in sporulation relating to ribosomal activity was first described as early as in the 1970s[18,19]. It was observed that the 30S ribosomal subunits differ between vegetative and sporulating cells in their ability to translate native mRNA, which is independent of 50S subunit, and that the mutations in 30S subunit and EF-G rendering the cells antibiotic-resistant also result in temperature-dependent

[1]Institute of Biochemistry and Biophysics, Polish Academy of Sciences, Pawinskiego 5a, Warsaw 02-106, Poland. [2]Maria Curie-Skłodowska University, Marii Curie-Skłodowskiej 5, Lublin 20-031, Poland. [3]The International Institute of Molecular Mechanisms and Machines Polish Academy of Sciences, M. Flisa 6, Warsaw 02-247, Poland. [4]These authors contributed equally: Olga Iwańska, Przemysław Latoch. ✉e-mail: agata.starosta@gmail.com

sporulation with blockage of sporulation at non-permissive temperatures[20–22]. Later, Ohashi et al. [23] identified a number of ribosomal proteins and translational factors required for sporulation but not affecting vegetative growth, including RpmA, RpmGB, RRF and EF-P, factor aiding translation of polyproline stretches[24,25]. The role of the latter was very recently confirmed by Feaga et al. [26] who, using ribosome profiling, showed that EF-P is important for Spo0A expression, and hence, sporulation initiation.

Here, we monitored ribosome location and activity, as well as translation profiles of *B. subtilis* WT 168 throughout the entire process of sporulation using confocal microscopy, click chemistry-based protein synthesis assays and ribosome profiling. We also examined translation in a triple knock-out mutant strain (3KO) lacking three paralogs of ribosomal proteins, RpmEB, RpmGC, RpsNB – zinc independent paralogs of zinc-binding ribosomal proteins RpmE (L31), RpmGA (L33) and RpsN (S14) respectively. Since entry into sporulation is dictated by nutrients deficiency, zinc limitation and the related ribosomal rearrangements may play an important role in protein synthesis regulation during sporulation[27]. In response to zinc deficiency, the two canonical ribosomal proteins RpmE and RpmGA are replaced by the zinc-independent RpmEB (L31*) and RpmGC (L33*) which increases intracellular zinc pool. RpsN is replaced by the RpsNB (S14*) later, to allow zinc-independent ribosome assembly de novo[28]. As the expression of these genes may be temporarily separated, using the 3KO mutant allowed us to monitor the effects of lack of protein substitution on translation during both early and late sporulation.

## Results

### The interplay between sigma factors regulates sporulation process

We performed ribosome profiling on the sporulating *B. subtilis*, covering the entire sporulation process, from sporulation induction to mother cell autolysis and spore release (eight samples T0–T7 in biological duplicates, grown and collected on two different days, RNA-seq and RIBO-seq performed in parallel; Supplementary Figs. 1 and 2). We investigated translation of genes belonging to six sigma regulons, (A, H, E, F, G and K), as well as selected sporulation regulons[29] (Supplementary Data 1). We employed k-means clustering to group highly translated genes into eight clusters based on their time-course translation level patterns (Fig. 1a, b). Clusterization revealed two main groups based on stage of sporulation (transition into sporulation/early sporulation and mid- to late sporulation), as well as functional grouping. In the early sporulation group, clusters 1 and 5 represent genes that are translated in the exponential growth and during transition into sporulation (peak at T0 and T1) and whose translation gradually decreases over time. The majority of genes in cluster 1 belongs to σ^A regulon and includes genes encoding biosynthesis of nucleotides (*pur* and *pyr* operons), carbon, amino acids and fatty acids metabolism, iron homeostasis and GTP and ATP synthesis, all of which represent metabolism under optimal growth conditions. The number of genes involved in regulation increases in cluster 5, which exhibits higher overall translation values compared to cluster 1. These regulatory genes include mostly sporulation initiation genes which fall into different regulons. The remaining genes in cluster 5 belong to σ^A regulon including mostly DNA replication, transcription and cell elongation and division genes. Cluster 3 includes genes that are highly translated throughout the entire sporulation process with an increase in translation between one and three hours post sporulation induction (T1–T3). High translation values of log10(TPM + 1) approximating 3 are due to a large subgroup of genes encoding ribosomal proteins (29 genes) and several translational factors, IF-1, RbfA, and *hpf* coding for ribosome hibernation promoting factor. These are accompanied by general stress response, stringent response and also competence genes. Genes in cluster 2 follow a similar pattern of translation, however, the overall log10(TPM + 1) values are one order of magnitude

lower and translation in the exponential phase is the lowest compared to the three remaining clusters in this clade. Cluster 2 encompasses the highest proportion of sporulation genes in its clade, including sporulation initiation, asymmetric septation, regulation of mother cell and forespore sigma factors activity, as well as sigma factors themselves (*bofA*, *fin*, *spoIIAA*, *spoIIAB*, *spoIIIL*, *sigF*, *sigE*, *sigG*), and genes from operons coding for toxins killing non-sporulating cells (*skf* and *sdp*, *spoIISB*).

The mid- to late sporulation clade also consists of four gene clusters dependent on the temporal activation and translation level. Clusters 6 and 7 represent genes with peak translation in mid-sporulation phase (T3 and T4) with similar temporal translation pattern but different overall translation value, which is one order of magnitude higher in cluster 7. Genes in cluster 7 represent mostly σ^E regulon and include genes responsible for initiation of spore coat assembly, the 'feeding tube' channel and a number of y-genes with unknown function. There are also a few genes belonging to the forespore sigma regulon, coding for transcription regulators. Cluster 6 genes represent both mother cell and forespore regulons. This cluster includes late sporulation mother cell sigma factor *sigK*, *sigKC* and *spoIVFA* which controls *sigK*. Genes in cluster 6 are responsible for spore and mother cell metabolism, including carbon, amino acid and fatty acid metabolism, spore maturation including dipicolinic acid uptake, and to a lesser extent spore coat and spore cortex biosynthesis as well as several germination related genes. Cluster 8 contains genes with the highest translation values and which are specific to late sporulation, with peak translation values in T5 and T6 (Fig. 1a). Genes in this cluster belong mostly to σ^G regulon, including essentially spore DNA protection proteins encoded by the *ssp* genes and a number of y-genes related to sporulation. A few genes assigned to cluster 8 from σ^K regulon are involved in spore coat and spore crust synthesis and germination (*gerPB*, *gerPD*). Genes segregated into cluster 4 represent a similar pattern of translation, however, the log10(TPM + 1) values are one order of magnitude lower. Vast majority of genes segregated into cluster 4 belong to σ^K regulon and include genes responsible for spore coat and spore crust synthesis (*cot* and *sps* genes), mother cell lysis genes, as well as a number of y-genes. Thus, clade with clusters 4 and 8 represents genes highly translated in late sporulation, regulated by mother cell and spore specific sigma factors respectively.

We observed the correlation of log2 transformed fold change of transcription and translation values between the consecutive time-points, focusing on the mother cell and forespore specific sigma regulons – σ^E/σ^K and σ^F/σ^G (Fig. 1c). The dynamics of transcription and translation of the mother cell and forespore regulon pairs correlated well, and a temporal interplay between the sigma factors can be seen. In the exponential growth and transition into sporulation (T0-T1), most genes from the mother cell and forespore regulons exhibited narrow fold change distribution in the range between −2.5 and 2.5 with the exception of *sigE*, *sigF*, *spoIIAA* and *spoIIGA*, with the fold change value of >5 in T1 (Fig. 1c). In consequence, transcription and even more so translation of these early sporulation regulons increased substantially in T2, exhibiting a broad range of fold change from around 0 to 10. Transcription and translation of both early sporulation regulons remained at a similar level or decreased slightly until the end of sporulation. The two late sporulation regulons, σ^G and σ^K, showed an increase in translation and transcription in T3 and T4 respectively and this continued until the T4 and T5. Interestingly, genes from both regulons exhibited higher fold change of translation than transcription compared to other CDSs. During late sporulation (T6-T7), translation and transcription of the σ^G and σ^K regulons did not change significantly as the fold change oscillated around 0. The rate of change of transcription and translation for all CDSs decreased gradually over the time course of sporulation with a distinct decline in the last two hours, as suggested by the decreasing trendline slope and R^2 values (Fig. 1c).

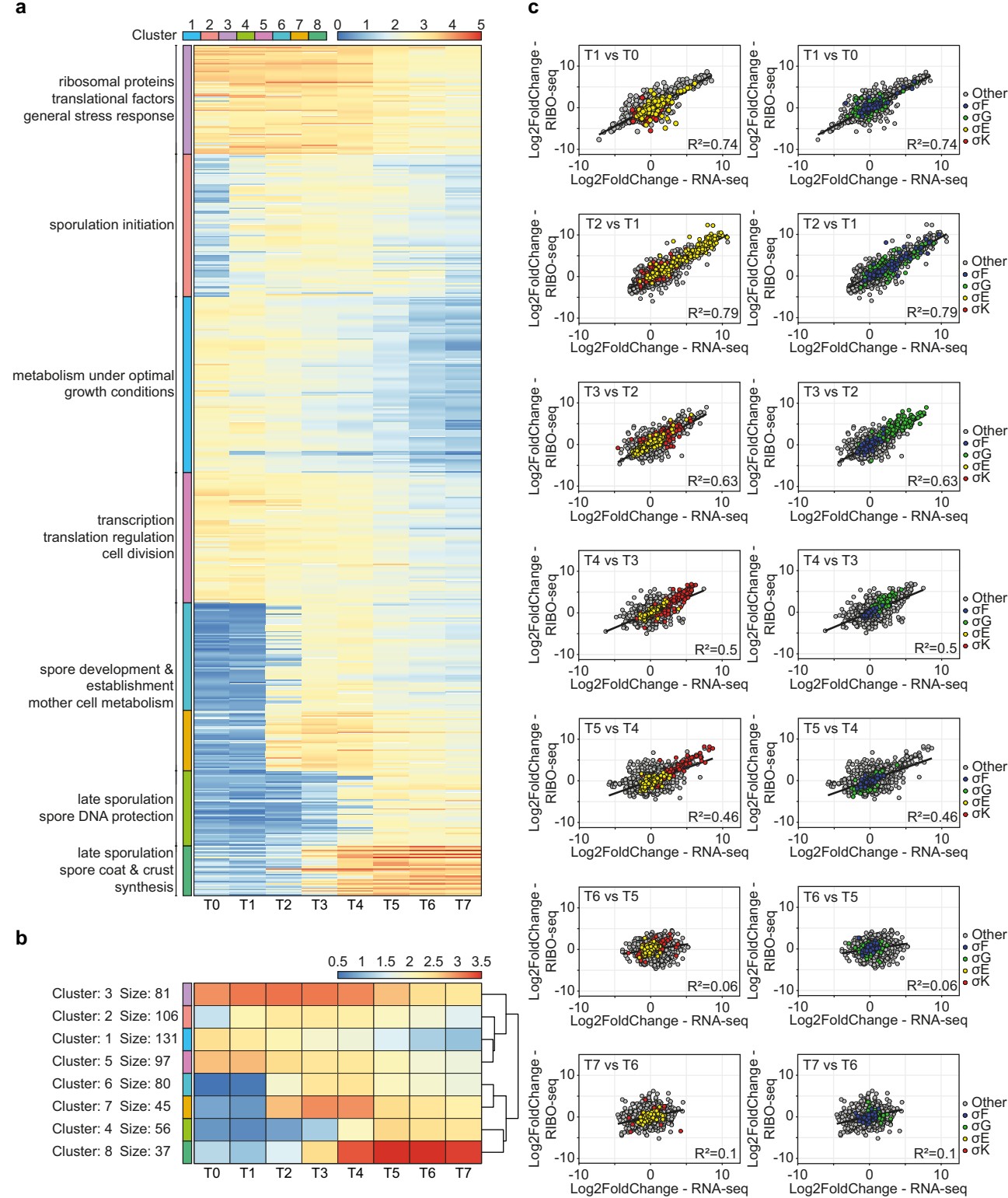

**Fig. 1 | The translational landscape of *B. subtilis* during sporulation reveals temporal clustering. a** Heatmap of the selected highly translated sporulation specific genes (mean values from two replicates of TPM ≥ 100, resulting in 633 genes). Highly translated genes from different sporulation regulons can be segregated into clusters of similar temporal activity and translation level based on k-means clustering. Data presented as log10(TPM + 1). **b** A summary heatmap of k-mean clusters constructed using Euclidean distance with complete linkage.

**c** Scatter plots illustrating correlation between log2 transformed values of fold change between transcription and translation at the consecutive timepoints during sporulation. Mother cell specific regulons are σ$^E$ in yellow and σ$^k$ in red; forespore specific sigma regulons are σ$^F$ in blue and σ$^G$ in green. Best fit lines and Pearson R2 values describe the dynamics and correlation of fold change for transcriptome and translatome data. Source data are provided as a Source Data file.

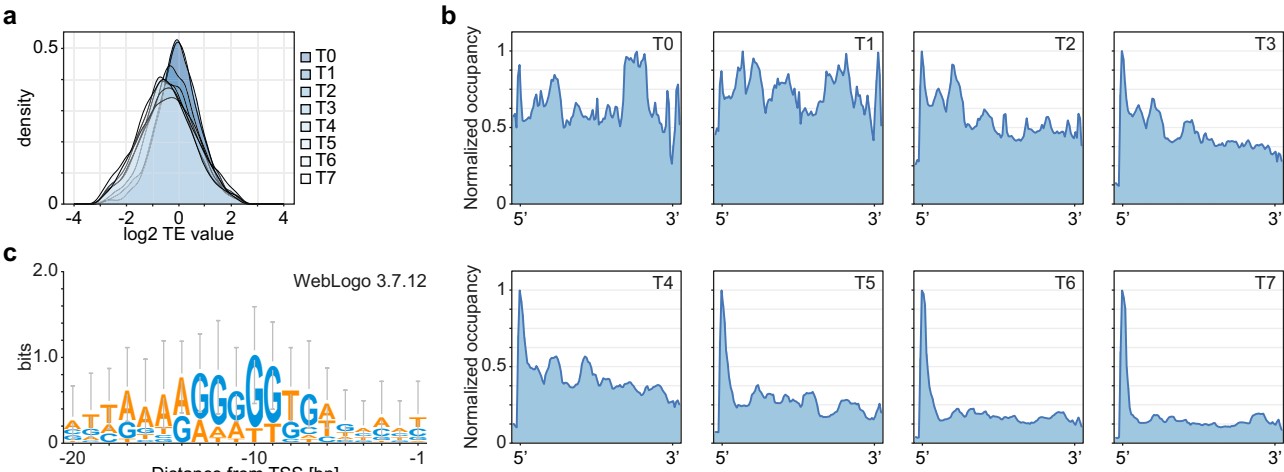

**Fig. 2 | The efficiency of translation gradually decreases during the process of sporulation in _Bacillus subtilis_. a** Histograms of translational efficiency of _B. subtilis_ during sporulation, calculated at T0 (prior to sporulation induction) to T7 (7 h post sporulation induction). **b** Mean ribosome density on CDSs with 5′ and 3′ UTRs (+/−50 nt) during sporulation in _B. subtilis_ normalized to the maximum peak value. **c** WebLogo representation of the sequence corresponding to the peak at 5′ UTR from timepoints T5–T7. Source data are provided as a Source Data file.

To verify whether translational efficiency (TE) follows a similar trend during sporulation, we plotted histograms of TE values for all CDSs at each timepoint (Fig. 2a), showing that TE also decreases gradually over time. We then investigated the mean ribosome occupancy on CDSs and 5′ and 3′ UTRs at each timepoint during sporulation. As expected, the mean ribosomal density on the CDSs was approximately uniform in exponential growth and early sporulation, and was then gradually decreasing, especially from T5 to T7. The peak corresponding to 5′ UTR increased significantly in these timepoints (Fig. 2b). The 5′ UTRs with high ribosome occupancy were shown to lie upstream of the late sporulation genes clustered into clusters 4 and 8 and belonging mostly to σ$^K$ and σ$^G$ regulons (Fig. 1a, Supplementary Data 1, Supplementary Fig. 3). We further investigated the sequences of the 5′ UTRs with the highest coverage of RPF in T5–T7 and demonstrated that these corresponded to the Shine–Dalgarno (SD) sequences in _B. subtilis_ (Fig. 2c). Very high ribosome coverage at the SD sequences, together with the lower overall TE in T6 and T7 suggest that the cell progressively silences translation in late sporulation.

## Localisation of translation during sporulation

We investigated the localisation of the translational machinery – the ribosome – during sporulation using fluorescent microscopy. We constructed WT-RpsB strain with GFP fused to the C-terminus of the RpsB protein (small subunit) which is encoded by the _rpsB_ gene. We observed the process of sporulation in one-hour intervals, beginning at T0 (before sporulation induction) up to T7 (seven hours post sporulation induction) (Fig. 3a, b). At T0, mid-exponential phase (OD600 - 0.6), the fluorescence is localised throughout the entire cell, with slight enrichment at the cell poles. The majority of ribosomes are located outside of the nucleoid, which was also reported by Lewis et al.[30]. At T1, the polar localisation of ribosomes is enhanced which can be attributed to cell shortening and chromosome condensation in the middle of the cell. Two hours post sporulation induction first asymmetric septa are being formed, followed by chromosome translocation. The ribosomes lose their polar localisation, which now becomes more uniform within the bacterial cell. However, the newly created forespore is almost devoid of the fluorescence signal and most of the ribosomes are located at the opposite cell pole and less at the asymmetric septum at the mother cell side (T3). This suggests a sequential packing of the forespore, with the chromosome being followed by the ribosomes. Indeed, in T4 and T5 we observed a gradually increasing GFP signal in the spore corresponding to the accumulation of

ribosomes in the forespore. The slight increase of the fluorescence at T6 in mother cell was observed which we hypothesise is a result of spore maturation and an increase in spore volume in relation to the mother cell leaving less space for the ribosomes to occupy. After seven hours of sporulation mother cell undergoes autolysis and a mature spore, packed with ribosomes, is released.

We investigated the localisation and intensity of translation in situ during the process of sporulation (T0–T6) using an alkyne analog of puromycin, o-propargyl-puromycin (OPP). Puromycin terminates translation by mimicking an aminoacyl-tRNA and binding to the nascent polypeptide chain. Following the addition of a picolyl azide fluorescent dye, a chemoselective reaction takes place ('click' chemistry reaction) which allows for visualisation and quantification of the incorporated OPP, or in other words, of the active translation[31] (Fig. 3c, d). We incubated _B. subtilis_ cells with OPP at one-hour intervals and labelled them with Alexa 488 dye. Interestingly, the active translation during exponential growth and one hour post sporulation induction was localised to the cell poles, away from the bacterial chromosome. This supports the recent findings of uncoupled transcription-translation in _B. subtilis_ and the runaway transcription model[17]. In the cells collected two hours post sporulation induction the fluorescent signal became more dispersed – following the ribosomal localisation – and reduced, pointing to an impeded translation. Once the asymmetric division took place, translation resumed and localised mostly to the septum (T3) and then predominantly to the forespore (T4). During late sporulation the fluorescent signal decreased, which was expected. After spore maturation, observed as the spore becoming phase bright, the spore is impermeable to the fluorescent dyes and non-specific labelling of the spore coat may be observed (beginning at T5), rendering data from later timepoints representative of the mother cell rather than the mature (phase bright) spore (Supplementary Fig. 4).

We investigated the levels and cellular fate of newly synthesised proteins during sporulation using a noncanonical methionine derivative, an azide-bearing azidohomoalanine (AHA), which was then chemoselectively tagged with a fluorescent dye using click-chemistry[32] (Fig. 3e). As incorporation of AHA does not terminate translation, the AHA-containing nascent chains can fold into proteins and reach their cellular localisations. The cells were incubated with AHA for 30 min at one-hour intervals beginning at T0 – sporulation induction – and fluorescently tagged using Alexa 488 dye. We then measured mean fluorescence intensity per cell in each timepoint, plotted the data to

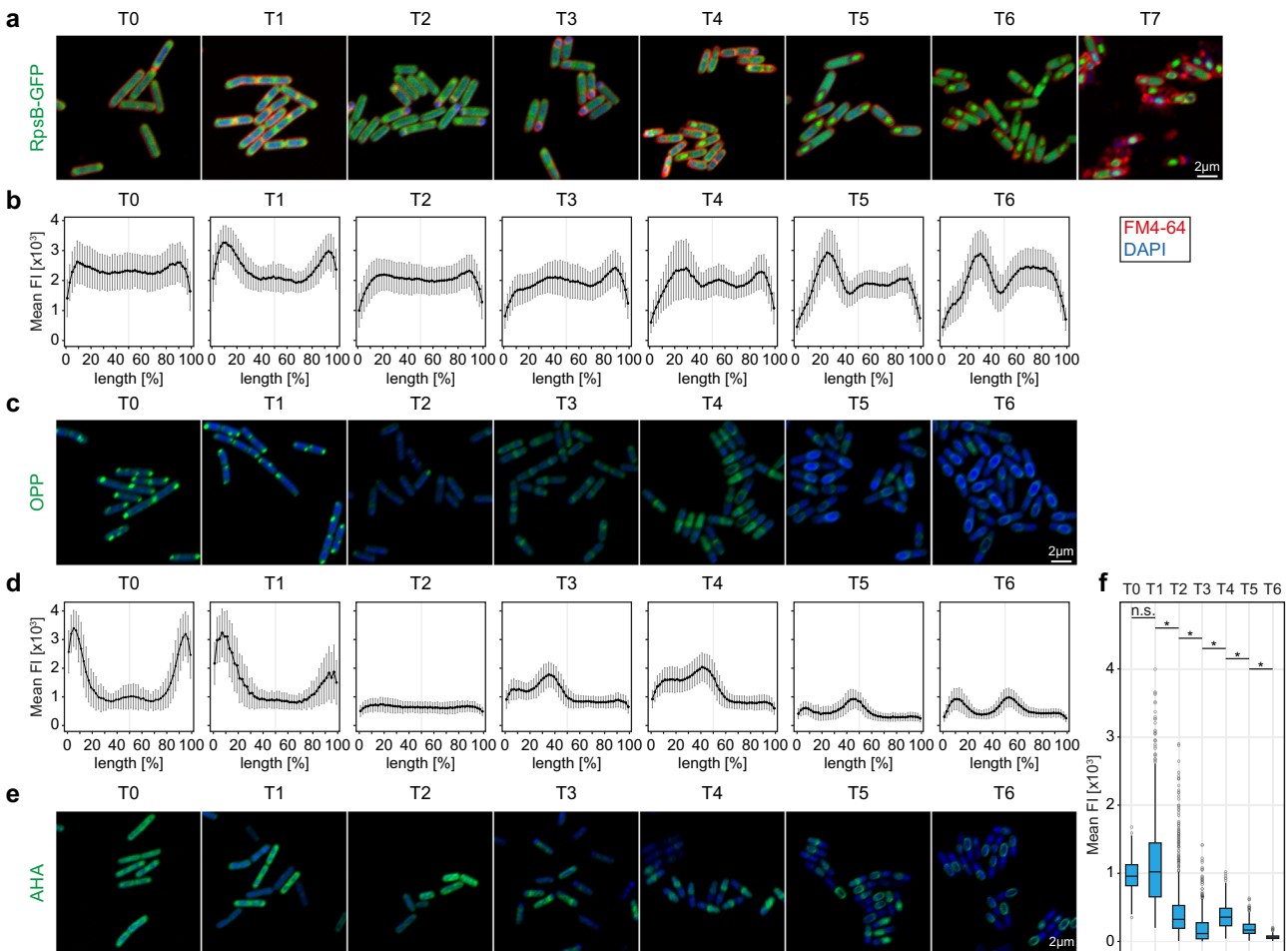

**Fig. 3 | Levels and localisation of translation during sporulation in *B. subtilis* are tightly controlled. a** Sporulation process of *B. subtilis* WT-RpsB-GFP cells. Cells were stained with DAPI to visualise the chromosome and SynaptoRed, a membrane stain, to track the asymmetric septation and sporulation progress. Samples were taken before sporulation induction (T0) and every hour of sporulation process (T1–T7). Scale bar is 2 μm. **b** Plots of mean values +/−SD of GFP fluorescence intensity across cells show subcellular localisation of ribosomes during sporulation. The cell lengths were normalised and fluorescence was measured along a line drawn across the cell long axis for the following numbers of cells: T0 – 109, T1 – 110, T2 – 110, T3 – 119, T4 – 116, T5 – 120, T6 – 120. **c** Microscopic images of *B. subtilis* WT cells treated with OPP and stained with Alexa 488 and DAPI, illustrating active translation during six hours of sporulation (T0–T6). Scale bar is 2 μm. **d** Plots of the mean valus +/−SD of the OPP-Alexa 488 fluorescence intensity across the cell showing localisation and intensity of active translation during sporulation. The cell lengths were normalised and fluorescence was measured along a line drawn across

the cell long axis, for the following numbers of cells: T0 – 226, T1 – 226, T2 – 226, T3 – 226, T4 – 226, T5 – 208, T6 – 226. **e** Microscopic images of *B. subtilis* WT cells treated with AHA and stained with Alexa 488 and DAPI, illustrating location of newly synthesized proteins throughout the process of sporulation (T0–T6). Scale bar is 2 μm. **f** Box and whiskers plots of mean AHA-Alexa 488 fluorescence intensity measured for *B. subtilis* WT cells during the sporulation process (*n*: T0 – 146, T1 – 280, T2 – 280, T3 – 280, T4 – 280, T5 – 280, T6 – 196). The plot illustrates fluctuations in the amount of newly synthesised proteins and changes in the population distribution at each timepoint during sporulation (two-sided Kruskal–Wallis, chi-squared = 1451.8, df = 6, *p*-value < 2.2e−16, asterisks indicate statistically significant pairs in post-hoc test [$p < 0.05$], T0 vs T1: *p*-value = 1; T1 vs T2: *p*-value = 2.03E−68; T2 vs T3: *p*-value = 1.08E−36; T3 vs T4: *p*-value = 4.47E−25; T4 vs T5: *p*-value = 6.03E−15; T5 vs T6: *p*-value = 1.09E−14). Centre line: median. Box: 25–75th percentiles (IQR). Whiskers: min/max values. Dots: outliers (1.5 * IQR). Source data are provided as a Source Data file.

investigate population changes and performed Kruskal–Wallis test with Dunn's post hoc tests with Bonferroni correction (Fig. 3f, Supplementary Data 2). During the exponential growth (T0) newly synthesised proteins are distributed approximately uniformly throughout the cell, with more pronounced fluoresce foci at the cell poles, which corresponds well with the localisation of ribosomes and active translation. As expected, the fluorescence intensity of the population follows normal distribution. Interestingly, one hour post sporulation induction, cellular fate of the newly synthesised proteins did not change significantly, however, the population became more heterogenous with a wider interquartile range and a number of outlier cells exhibiting high mean fluorescence. This is most probably a result of proteome adaptation to the conditions of nutrient limitation involving activation of several different nutrient sequestration pathways before the cell commits to sporulation[33]. The increased intensities may

perhaps also be attributed, in a small degree, to the cells experiencing starvation resulting in an increased AHA transport into the cell. In this transition phase (T1–T2), a smaller subpopulation of cells has an increased fluorescence signal, while the rest slowly suppresses protein synthesis in preparation for sporulation. This trend continues to three hours post sporulation induction with a decreasing interquartile range suggesting that the population becomes less heterogenous. In T3 the majority of newly synthesised proteins are located at the asymmetric septum which correlates well with the active translation and cellular localisation of the ribosomes. At this time newly synthesised proteins localise mostly to the mother cell which changes four hours post sporulation induction, when the ribosomes become present in the forespore at a higher density. At T4 we observed an upshift in the mean fluorescence intensity, as well as more symmetrical distribution of the mean fluorescence with newly synthesised proteins localised

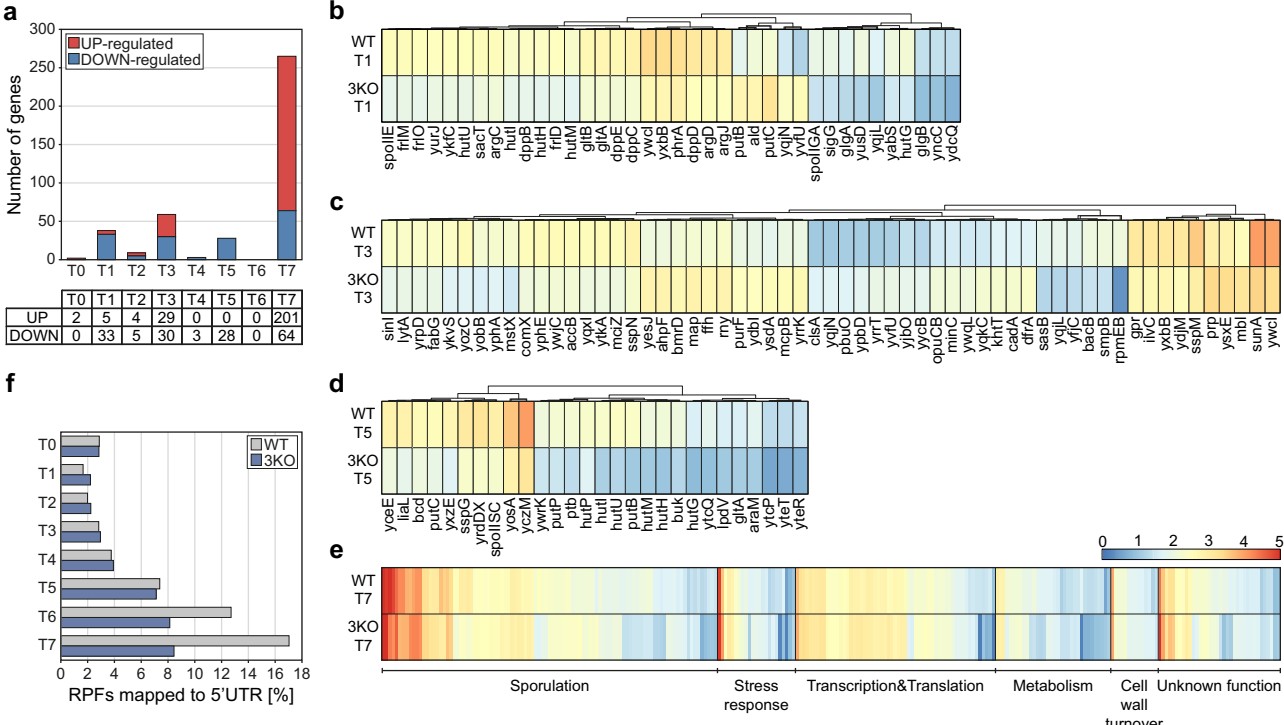

**Fig. 4 | Differences in the translatome (RIBO-seq) between WT and 3KO mutant.** **a** Bar plot and table displaying the number of statistically significant genes from the differential expression analysis for all time points (T0–T7), divided into UP-regulated and DOWN-regulated genes in 3KO. Heatmaps of log10(TPM + 1) values for statistically significant genes at: **b** 1; **c** 3; **d** 5; **e** 7 h post-sporulation induction. All heatmaps use the same scale as presented in the panel **e**. Dendrograms represent hierarchical clustering using Euclidean distance with complete linkage. Statistically significant genes, as identified with DESeq2, were selected based on padj ≤ 0.05 and |log2FoldChange|>0.6. A two-sided Wald test was used to determine statistical significance, with *p*-values adjusted for multiple comparisons using the Benjamini–Hochberg method. **f** Bar plot depicting the percentage of RPFs (ribosome-protected fragments) mapped to the 5'UTR for WT and 3KO across all time points (T0–T7).

predominantly at the asymmetric septum and in the forespore. At T5 and T6 the mean fluorescence intensity gradually decreases indicating reduced protein synthesis in the mother cell.

**Paralogs of ribosomal proteins during sporulation**

We constructed a triple knockout strain (3KO) carrying deletions of genes encoding paralogs of ribosomal proteins L31, L33 and S14 – RpmEB, RpmGC, RpsNB respectively – lacking a CXXC zinc binding motif (Supplementary Fig. 5). Although *rpmGC* is a pseudogene in 168 lineage[34], we decided to knock out all zinc-independent paralogs. We confirmed the ribosomal localisation of RpmEB microscopically by tagging the protein with mCherry in a native locus and tracking the ribosome localisation during sporulation (Supplementary Fig. 6). The ribosomal localisation of RpmEB (untagged) was also verified by mass spectrometry. The fractions containing 70S ribosomes were purified by sucrose density gradient centrifugation from WT and 3KO strains and their protein composition was analysed (Supplementary Data 3). Low expression levels, occlusive position on the ribosome and high similarity between the paralog and the canonical form of the analysed peptides in MS prohibited us from making similar observations for the remaining two paralogs. The existing literature however reports ribosomal localisation of both paralogs[28,35].

The 3KO strain demonstrated normal morphology and growth in CH medium at 37 °C compared to WT (Supplementary Fig. 7). However, sporulation efficiency based on the microscopic observations was delayed up to four hours post sporulation induction, after which time it reached the values of WT (Supplementary Table 1a). The sporulation/germination efficiency measured as a ratio of CFU resulting from fully sporulated cultures before and after heat treatment (40 min at 90 °C) showed a 50% decrease in spore formation/

germination compared to WT (Supplementary Table 1b) suggesting a possibly aberrant spore formation in the 3KO strain. To investigate this, we sequenced transcriptome and translatome of the 3KO mutant during sporulation, under the same conditions as for WT – from T0 (exponential growth/sporulation induction) to T7 (7 h after sporulation induction), in biological duplicates (Supplementary Figs. 1 and 2). We examined differentially expressed genes (DEGs) between the 3KO and WT in all timepoints, based on the translatome data (Supplementary Data 4). As expected, in exponential growth (T0) there were no significant differences in expression. Interestingly, in timepoints T2, T4 and T6, there were little or no DEGs (Fig. 4a). However, one hour post sporulation induction 30 genes were downregulated in 3KO (p adj. < 0.05) and these fell into two main groups: metabolism and sporulation regulation genes. Importantly, *spoIIGA* and *spoIIE* responsible for maturation and control of early sporulation sigma factors σ^E and σ^F, as well as *sigG*, were downregulated in 3KO. Amino acids, amines and dipeptides metabolism including uptake, synthesis and utilisation, and glycogen synthesis under σ^E control were also downregulated in 3KO (Fig. 4b). In T3, the number of DEGs in 3KO increased twice. The affected cellular processes included downregulation of stress response related to sporulation, including cannibalism, cell division and alarmone synthesis. On the other hand, the upregulated genes included processing of ribosome, mRNA, rRNA, and cell wall and cell membrane morphology and transport among others (Fig. 4c), suggesting a delay in entry into sporulation. In T5, similarly to T1, 30 genes were downregulated in 3KO and again these could be categorized into nitrogen and carbon metabolism as well as stress/sporulation groups. Utilisation of amino acids proline and histidine, as well as branched-chain keto-acids was downregulated, together with transport and utilization of carbon sources. Sporulation and/or stress

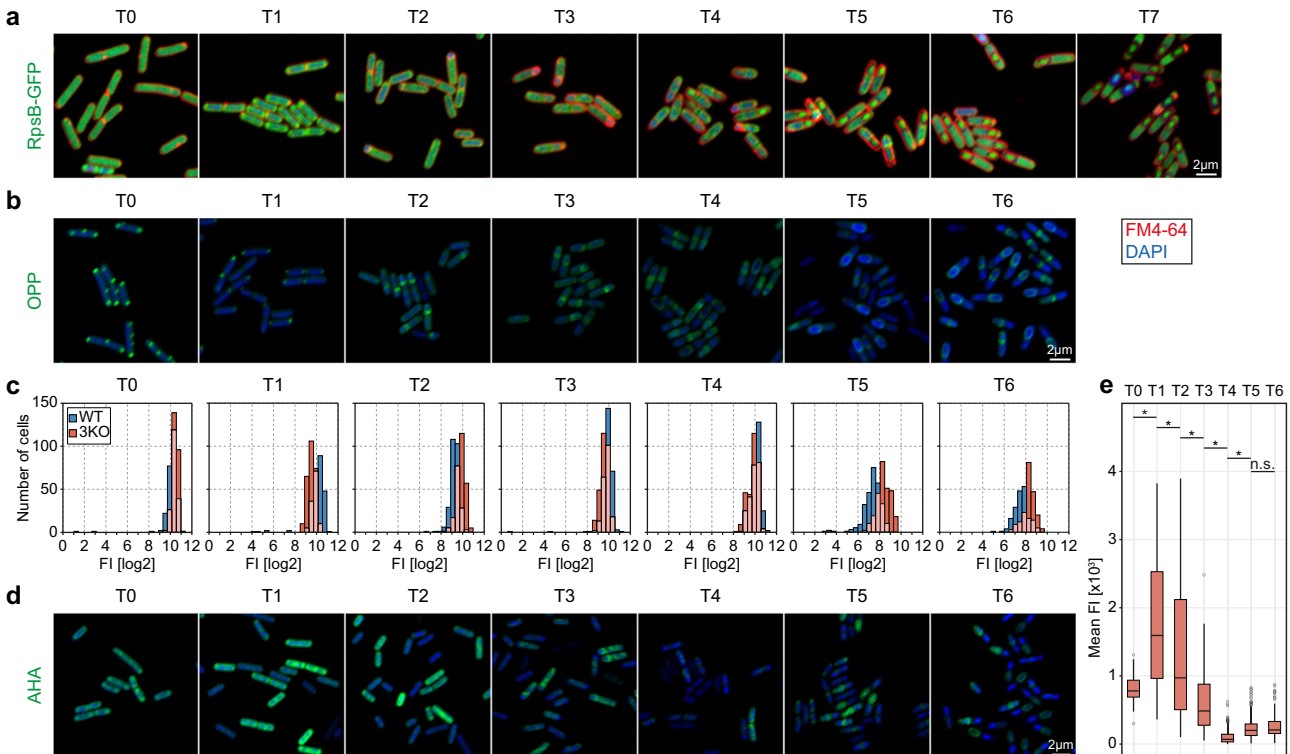

**Fig. 5 | Translation in the 3KO mutant is dysregulated in late sporulation.**
**a** Sporulation process of *B. subtilis* 3KO-RpsB-GFP cells. Cells were stained with DAPI to visualise the chromosome and SynaptoRed to track the asymmetric septation and sporulation progress. Samples were taken before sporulation induction (T0) and then hourly until T7, 7 h post sporulation induction. Images of representative cells obtained from two biological replicates. Scale bar is 2 µm.
**b** Microscopic images of 3KO cells treated with OPP and stained with Alexa 488 and DAPI, illustrating active translation during six hours of sporulation (T0–T6). Images of representative cells obtained from two biological replicates. Scale bar is 2 µm.
**c** Histograms of mean log2 transformed fluorescence intensity of OPP-Alexa 488 in WT (blue) and 3KO (orange) populations during sporulation. In late sporulation mean fluorescence is higher in 3KO and in T6 two subpopulations of the 3KO strain emerge. The overlapping portions of histograms are in light orange. The cell lengths were normalised and fluorescence was measured along a line drawn across the cell long axis. **d** Microscopic images of 3KO cells treated with AHA and stained with Alexa 488 and DAPI, illustrating location of newly synthesized proteins throughout the process of sporulation (T0–T6). Scale bar is 2 µm. **e** Box and whiskers plots of mean AHA-Alexa 488 fluorescence intensity measured for 3KO cells during the sporulation process (*n*: T0 – 121, T1 – 206, T2 – 260, T3 – 260, T4 – 260, T5 – 260, T6 – 260). The plot illustrates fluctuations in the amount of newly synthesised proteins and changes in the population distribution at each timepoint during sporulation (two-sided Kruskal–Wallis, chi-squared = 1314.6, df = 6, *p*-value < 2.2e−16, asterisks indicate statistically significant pairs in post-hoc test [*p* < 0.05], T0 vs T1: *p*-value = 0.010706683; T1 vs T2: *p*-value = 0.006432589; T2 vs T3: *p*-value = 7.83E−10; T3 vs T4: *p*-value = 2.79E−79; T4 vs T5: *p*-value = 2.04E−17; T5 vs T6: *p*-value = 1). Centre line: median. Box: 25–75th percentiles (IQR). Whiskers: min/max values. Dots: outliers (1.5 * IQR). Source data are provided as a Source Data file.

related genes included several y-genes, a spore DNA protection protein, toxins and resistance encoding genes, pointing to a dysregulated metabolism and stress response (Fig. 4d). The most pronounced difference in translation, based on DEGs, was observed in T7 with over 50 downregulated genes and more than 200 genes exhibiting higher translation, compared to WT. In both up- and downregulated sets of genes we found metabolism, sporulation, germination and stress response related genes, suggesting that these processes are highly dysregulated in the 3KO mutant in late sporulation. Especially, we identified a group of genes specifically upregulated in 3KO – translation related genes including 32 ribosomal protein coding genes and transcription and translation regulation genes (Fig. 4e). Also, peptidoglycan synthesis and turnover genes, as well as a large number of spore coat and spore crust genes were significantly upregulated, with the spore DNA protection proteins (*ssp* genes) downregulated in 3KO in T7. The translatome data suggests delayed sporulation in the early stages and dysregulated translation and metabolism in late sporulation. Moreover, upregulation of ribosomal and transcription and translation regulation genes may suggest that the 3KO spores are not as efficient in silencing their gene expression regulation in preparation for dormancy as the WT spores.

To investigate this further, we examined the average footprint coverage on the CDSs and 5′ and 3′ UTRs in the 3KO strain and found that there is a smaller percentage of footprints mapped to 5′UTRs compared to WT in late sporulation (Fig. 4f). Interestingly, this disproportion was mostly due to a high amount of reads localised to 5′UTR of *yhcV* gene in WT, encoding a forespore specific late sporulation protein. Expression of this gene in late sporulation was also recorded by others, however its function remains unknown[36].

Microscopic observations of the sporulating 3KO strain were performed every hour from T0 (exponential growth) to T7 (seven hours post sporulation induction) using fluorescent microscopy. As for WT, we constructed the 3KO-RpsB-GFP strain with GFP fused to the C terminal of the RpsB protein in the 3KO background. To better visualise the progress of sporulation, the 3KO-RpsB-GFP cells were stained with DAPI and SynaptoRed (Fig. 5a). Localisation of the ribosome during sporulation in the 3KO-RpsB-GFP strain is very similar to the WT-RpsB-GFP strain. Briefly, during the exponential growth ribosomal localisation is rather uniform with the ribosomes segregating to the cell poles at T1 and then, to the asymmetric septum at the mother cell side. After the forespore engulfment around 4 h post sporulation induction, the fluorescence signal is gradually increasing in the spore suggesting ribosomes packing and spore maturation. However, as mentioned above, asymmetric septation in the 3KO strain is delayed with less forespores being formed by T3 compared to WT-RpsB-GFP. Also, we noticed that although spore morphology appears normal

under the confocal microscope, much less spores are being released by the mother cells, which seem to not undergo autolysis as efficiently as the WT cells, or the autolysis is much delayed.

We then investigated active translation during sporulation (T0–T6) in situ in the 3KO strain using puromycin analog, OPP (Fig. 5b). We measured mean fluorescence intensities per cell compared these with the WT values (Fig. 5c) and showed that translation was gradually decreasing throughout the sporulation process. Both the localisation of active translation and fluorescence intensities between the 3KO and WT strains were very similar until T5. Five and six hours post sporulation induction however, the distributions of mean fluorescence intensity in 3KO were significantly different to WT (Kolmogorov–Smirnoff, WT vs 3KO at T5: D = 0.55314, $p < 2.2e{-}16$; WT vs 3KO at T6: D = 0.56854, $p < 2.2e{-}16$). 3KO showed higher mean fluorescence which suggests dysregulated translation, specifically disrupted translation silencing in the mother cell. This stays in agreement with our ribosome profiling data. To investigate how this relates to the cellular fates and amounts of newly synthesised peptides in 3KO, we used the methionine derivative AHA (Fig. 5d, e) and performed Kruskal–Wallis test with Dunn's post hoc tests with Bonferroni correction (Supplementary Data 2). As expected, during exponential growth the localisation and distribution of fluorescence intensities were very similar to WT, following normal distribution. One hour post sporulation induction the population became very heterogenous with increased mean fluorescence compared to WT. The considerable heterogeneity and higher mean values in comparison to WT continued in T2 and T3, with the mean fluorescence intensities gradually decreasing until 4 h post sporulation induction, when the population reached minimum mean fluorescence intensity values with a narrow interquartile rage. In T5 and T6 the amount of newly synthesised proteins however began to increase, unlike in WT, which points towards imbalanced translation regulation in late sporulation and correlates with ribosome profiling data and other microscopic observations presented here.

## Discussion

Translation is the most metabolically demanding process in a bacterial cell[37] and as such must be tightly regulated upon major metabolic and morphological changes such as sporulation. Here, we show that translation in *B. subtilis* is organized both temporally and spatially during sporulation and that translational efficiency gradually decreases throughout the process. Our results complete the published transcriptome data[4,36], by adding an extra layer of information regarding *B. subtilis* translatome throughout sporulation.

Analysis of translation dynamics of genes involved in sporulation revealed distinct temporal patterns associated with functional categories at different stages of sporulation. The most highly translated CDSs included ribosomal genes and translation factors during logarithmic and early sporulation stages (cluster 3, T0–T4) and the *ssp* genes with a number of y-genes related to late sporulation (cluster 8, T4–T7). Although upon entry into quiescence and in stringent response translation of the *rrn* genes and rRNA synthesis is inhibited[16,38], the presence of steady state levels of translational machinery has been reported in stringent response[39] and in sporulation, until forespore engulfment[40]. However, translation does decrease gradually throughout the process of sporulation which can be seen from the decreasing TE values and ribosomal pausing at SD sites in late sporulation. Since translation initiation is often a rate limiting factor in translation, the accumulation of RPFs at SD sequence and start codons suggests prolonged initiation times and therefore lower translation levels[41]. Very high expression values of the late sporulation genes in cluster 8 can be assigned to the spore σ<sup>G</sup> regulon, which points to a high translational activity of the spore. Indeed, although the overall translation decreases, data presented here suggests that spores should not be viewed as translationally inactive throughout the process of

sporulation. In fact, more evidence is emerging indicating spore transcriptional and translation activity during periods of supposed inertness[13,40,42].

Using fluorescence microscopy, we showed in situ that translation and translational machinery are subcellularly localised in *B. subtilis* during sporulation. Data collected here stay in agreement with the seminal paper by Johnson et al.[17] on uncoupled transcription and translation in *B. subtilis*. As shown above, translation during logarithmic growth localizes to the cell poles at some distance away from the centrally located chromosome, where transcription takes place[30]. Ribosomal subunits are diffused in the cytoplasm and assemble at sites of active translation[39], which is demonstrated by differences if the fluorescence intensity plots of RpsB-GFP and Alexa-labelled OPP-arrested ribosomes. In response to nutrient limitation (T1) the bacterial population becomes heterogenous in terms of nascent peptide synthesis. This illustrates the bimodal differentiation during sporulation induction – a regulatory mechanisms in which two populations emerge in anticipation for a change in the environmental conditions[43]. Interestingly, at T1 we also noticed a decrease in the active translation levels at one of the cell poles. In light of the recent research on the localisation and mechanism of the asymmetric division machinery[44,45], we wonder whether such decrease in translation activity might possibly be in preparation for the asymmetric septation at a selected cell pole. Based on the OPP assay we were also able to identify two events of translation silencing in the mother cell – immediately prior to the asymmetric septation (T2) and at the final stages of sporulation (T6). This corresponds well with the levels of nascent peptide synthesis which decrease at T3 resulting in a heterogenous population, and then in T6 when the decrease is more evident and the population is more homogenous. Interestingly, translation silencing at T2 was accompanied by the loss of polar localisation of ribosomes. We hypothesise that such loss of localisation and transient translational halt, together with asymmetric septation and chromosome translocation, are necessary steps during sporulation initiation.

Zinc is a required metal as it is a cofactor for many essential proteins and its intracellular levels are tightly regulated in *B. subtilis*. Three paralogous ribosomal proteins, RpmEB, RpmGC and RpsNB, take part in the zinc starvation response by mobilizing zinc from ribosomes or allowing zinc-independent ribosome assembly[34]. However, a regulatory role of such zinc depleted ribosomes in sporulation has been understudied[27]. The 3KO strain showed extended times of sporulation initiation and delayed germination, although a similar number of spores was produced compared to WT. This stays in agreement with Mutlu et al.[46], who reported that a delay in sporulation timing has negative effects on the spore ability for successful revival. It should also be noted however that the 3KO strain showed decreased mother cell lysis which could potentially results in prolonged germination times due to less spores being released. Based on the translatome data, we showed that the effects of the triple deletion of the zinc-independent paralogs were pleiotropic during sporulation in *B. subtilis*. In early sporulation the most affected cellular functions were downregulated sporulation with decreased levels of *spoIIE* and *spoIIGA*, stress response and metabolism, suggesting an extended period of bimodal differentiation in 3KO. Such dysregulated cell response continued up to late sporulation, with most pronounced difference at T7. Although zinc can be linked to the C-1 metabolism including synthesis of purines, serine, glycine, methionine and formylmethionyl-tRNA which also affect translation[47], these pathways did not differ significantly in our dataset. Zinc limitation as a common factor behind the observed effects cannot be definitely eliminated here however, we find this rather unlikely as none of the zur regulon genes were significantly upregulated in 3KO[28]. Rather, translation of peptidoglycan synthesis and turnover, metabolism and ribosomal proteins was increased in T7, which does not point towards zinc deprivation conditions. Therefore, we propose that the lack of the subpopulation of ribosomes carrying

zinc independent paralogs of ribosomal proteins results in the cell's inability to adequately silence translation in preparation for dormancy which appears to be essential for proper spore formation and germination. This may indicate ribosomal finetuning or a regulatory role of a subpopulation of zinc depleted ribosomes during sporulation and growth arrest in *B. subtilis*. However, more data including in vitro experiments is needed to prove this.

We confirmed the extended bimodal differentiation and the resulting populational heterogeneity of translation levels in 3KO cells in situ. The 3KO strain also showed discrepancies in the translational silencing pattern that we identified in WT. The lack of zinc-independent ribosomal proteins may therefore disrupt the adaptation of translational machinery in preparation for dormancy and affect the ribosome dynamics. Although the mechanisms used by the cell to adapt to translation silencing during sporulation and growth arrest are not well understood[48], the change in protein composition of the ribosome may play a substantial role.

Post-transcriptional control of gene expression and the regulatory role of ribosomes in translation especially in the conditions of growth arrest and quiescence are still underexplored[48]. Here we showed that the *B. subtilis* cell entering dormancy upon nutrient limitation remains levels of translational activity and moreover, precisely controls them spatially and temporally during the process of sporulation.

## Methods

### Strains and growth conditions

*B. subtilis* 168 strain was used as a wild type and the remaining strains are derivatives thereof, as listed in Supplementary Table 2a. The strains were grown overnight in LB medium at 30 °C with shaking then diluted to $OD_{600} = 0.1$ in CH medium[49] and grown until $OD_{600}$ reached 0.5–0.6, at 37 °C with shaking. Sporulation was induced by medium exchange to sporulation medium as described by Sterlini and Mandelstam (1968)[50], except the cells were harvested by filtration and filters were transferred into the culture flasks. The sporulation medium was supplemented with 3% v/v of the culture in the CH medium at $OD_{600} = 0.5$–0.6 to promote sporulation.

### Strains construction

The list of primers and plasmids used in this study are listed in Supplementary Table 2b and c, respectively.

To construct the 3KO mutant, competent cells of *B. subtilis* strains were transformed using standard techniques[51] with linear ~5 kb DNA constructs prepared by overlap PCR method[52]. The knock-out mutations were performed sequentially by introducing one KO at a time. The DNA constructs were prepared by PCR amplification and then fusion of the following: approximately 2 kb genomic fragments upstream and downstream of the gene of interest were amplified from *B. subtilis* chromosomal DNA using primers UPFOR/UPREV and DOWNFOR/DOWNREV, separate for each gene; an antibiotic cassette was amplified by PCR from an appropriate plasmid using primer pair MIDFOR/MIDREV, specific for each gene (Supplementary Table 2b, c). PCR amplification of the resistance cassette introduced flanking regions of the genomic locus of the gene of interest to both sides of the cassette (Supplementary Table 2c). The fragments were then fused in a secondary PCR amplification resulting in an approximately 5 kb linear DNA product. In the triple knock-out strain the paralogs of ribosomal protein genes *rpmEB* (BSU_30700), *rpsNB* (BSU_08880) and *rpmGC* (new_2477758_2477958_c) were replaced with kanamycin, chloramphenicol and erythromycin antibiotic cassettes respectively and the transformants were selected on nutrient agar plates with appropriate selection. The results of transformation were verified by PCR and Sanger sequencing of the relevant genomic regions.

In the WT-RpsB-GFP and 3KO-RpsB-GFP strains, the RpsB ribosomal protein was tagged with GFP at the C-terminus. The fusion was performed by a double cross-over and integrated into the chromosome in the native locus. The tag was introduced by transforming the competent *B. subtilis* cells[51] with a linear DNA construct prepared by overlap PCR method[52]. The 5.6 kb DNA construct consisted of two genomic regions 2 kb upstream and downstream of the STOP codon of *rpsB* gene amplified from the genomic DNA of *B. subtilis* using UPFOR/UPREV and DOWNFOR/DOWNREV primers, and gfp tag and spectinomycin resistance cassette were amplified from pSHP2 plasmid with MIDFOR/MIDREV primers introducing appropriate flanking regions (Supplementary Table 2b, c). The STOP codon of *rpsB* was omitted. The transformants were selected on nutrient agar plates with spectinomycin and the results of transformations were verified by PCR and visually (expression of GFP). The WT-RpsB-GFP-RpmEB-mCherry strain with fluorescently tagged RpmEB with mCherry at the C terminus was constructed analogously. The competent WT-RpsB-GFP strain was transformed with a linear construct carrying the mCherry gene and chloramphenicol resistance cassette from the pMCL200 plasmid (Supplementary Table 2b, c). The fusion was performed by a double cross-over in the native locus (*rpmEB*).

### Sporulation/Germination efficiency assay

WT and 3KO strains were grown and induced to sporulate as described above. Spores were left to mature overnight at 37 °C with shaking. Spore suspensions were diluted to $OD_{600} = 1.5$ and divided into two aliquotes of 250 ul. One aliquot was heat treated for 40 min at 90 °C and the second was not (control). Three dilutions of heat treated and control aliquots were streaked on nutrient agar plates and incubated overnight at 30 °C. Sporulation/germination efficiency was calculated as (CFU heat treated/CFU control)*100%. This was done in triplicate for each strain and mean value was calculated.

### RNA isolation, library preparation and sequencing

RNA was isolated from sporulating WT and 3KO strains. Cultures were harvested hourly for seven hours into sporulation, beginning at T0 (prior to sporulation induction), resulting in a total of eight timepoints. Before harvesting, cultures were treated with 0.3 mM chloramphenicol. Cells were collected by filtration and flash frozen in liquid nitrogen. Purification of mRNA, ribosomal footprint isolation and library preparation was performed as described earlier[53]. Briefly, frozen pellets were ground using mortar and pestle with the addition of aluminium oxide and lysis buffer (20 mM TRIS pH 7.6, 10 mM MgAcet, 150 mM KAcet, 0.4% TRITON X-100, 6 mM β-mercaptoethanol, 5 mM CaCl2, 1 mM chloramphenicol, with 2 mM GMP-PNP and 100 U/mL of DNAse I). The extracted total RNA was used for purification of mRNA and ribosomal footprint isolation. rRNA depletion was performed with Invitrogen Microbexpress kit to yield mRNA which was then fragmented by alkaline hydrolysis (2x alkaline hydrolysis buffer: 220 µL of 0.1 M NaHCO3, 30 µL of 0.1 M Na2CO3 and 1 µL of 0.5 M EDTA, samples were incubated at 95 °C for 25 minutes). Ribosomal footprint isolation was performed by MNase digestion of the total RNA (per 1 mg of RNA, 3.8 µL of 187.5 U/µL Mnase in 10 mM Tris pH 8 was added and incubated at 25 °C, for 45 min with mixing, 300 RPM) followed by size selection using polyacrylamide gel electrophoresis. The obtained mRNA fragments and ribosomal footprints were end-repaired (dephosphorylation of 3′ ends at 37 °C for 1.5 h, followed by phosphorylation of 5′ ends at 37 °C for 1 h in the presence of ATP, using T4 PNK) and used for preparation of cDNA sequencing libraries with NEBNext Multiplex Small RNA Library Prep Set for Illumina, according to the manufacturer's guidelines. Sequencing was performed on an Illumina NextSeq500 by Genomed S.A. (Warsaw, Poland) and Genome Facility at CeNT (Warsaw, Poland); single-end 50 bp reads were sequenced.

### Sequencing data analysis

FastQC[54] reports were utilized to assess raw data quality. Following this step, the removal of adaptors and low-quality sequences was

performed using TrimGalore![55] software. The Bowtie[56] tool was employed to eliminate sequences corresponding to tRNA and rRNA using genome reference data (RefSeq: NC_000964.3). For RIBO-seq data, filtering was applied, retaining only sequences between 15-32 nucleotides in length. After preprocessing, the FastQC[54] tool was used for quality control. Length distribution plots were created to analyze the read length distribution of the RIBO-seq data. The clean and trimmed data was mapped to the genome using STAR[57], with a tolerance of up to 4% mismatch in read length and exclusion of splicing. The resulting BAM files were sorted and indexed using the sambamba[58] software. Uniquely mapped reads were counted using featureCounts[59], employing gene and UTR annotations from the BSGatlas[60] database, with a minimum overlap of 4 nucleotides. To produce comprehensive statistics, tables summarizing the trimming, filtering, mapping, and counting steps were created (Supplementary Data 5). Subsequently, the normalization of read counts was carried out using the Transcripts Per Million (TPM) method for CDS, UTR, and combined data. Spearman correlations were calculated between biological replicates at all time points, using the cor function from the standard R stats package. Translation efficiency was determined by dividing normalized RIBO-seq data by RNA-seq data for all coding sequence regions. Heatmaps were created using normalized RIBO-seq data, focusing on sporulation genes (Supplementary Data 1) and sigma factor regulons (sigA, sigE, sigF, sigG, sigH, and sigK). Highly expressed genes were selected based on a threshold of average TPM values across all time points, set at 100. The K-means clustering algorithm was utilized to cluster into a pre-defined number of groups (8). For RIBO-seq data, BAM files were converted to BED format and subjected to metagene analysis with STATR[61] tool. Metagene plots were generated for footprint lengths of 20 and 24 nucleotides. Furthermore, average coverage plots for all genes and clusters of genes were produced using the DeepTools[62] library using BPM normalization (bins per million mapped reads). Principal Component Analysis (PCA) was conducted, and normalized data was used to generate plots to explore structure and variability. Differential gene expression analysis was carried out on RNA-seq and RIBO-seq data using the DEBrowser[63] tool and the DESeq2[64] library. Correlation plots of log2FoldChange between experiments were produced, with sigma factor regulons obtained from SubtiWiki[29] highlighted for reference. Pearson $R^2$ values were obtained using the cor function from the standard R stats package.

## Confocal microscopy

Confocal microscopy was carried out with an inverted confocal system Nikon C1. Images were taken with Plan-Apo VC 100×/1.40 oil immersion objective. GFP/Alexa 488 were excited at 488 nm, DAPI at 408 nm, SynaptoRed at 543 nm. The fluorescence signals were collected using filter sets: 515/30 nm for GFP/Alexa 488, 480/40 nm for DAPI and 610LP nm for SynaptoRed. Imaging of specimens having more than one fluorophore was performed in a sequential scan mode to prevent bleed-through of signal. *B. subtilis* strains were grown and sporulation was induced as described above. Aliquots of the cultures were sampled every hour for seven hours into sporulation, beginning at time $T_0$ – prior to sporulation induction. Cells were immobilised on a glass microscope slides covered with a thin film of 1% agarose and a cover slip. Unless stated otherwise, the cells were visualised with SyanptoRed dye at a final concentration of 10 µg ml$^{-1}$ and 4′,6-diamidino-2-phenylindole (DAPI) dye at a final concentration of 5 µg ml$^{-1}$. For translation visualisation, cells were incubated with O-propargyl-puromycin (OPP) using Click-iT® Plus OPP Alexa Fluor® 488 Protein Synthesis Assay Kit (Invitrogen) according to the manufacturer's guidelines. Briefly, cells we incubated with 13 µM OPP for 30 min at 37 °C with shaking. Cells were then fixed with 3.7% formaldehyde and permeabilized with 0.1% Triton X-100. Fluorescent labelling was performed for 30 min with Alexa Fluor 488 reaction cocktail. For visualisation of protein synthesis, Click-iT® AHA Alexa Fluor® 488 Protein Synthesis HCS Assay (Invitrogen) kit was used. Cells were incubated with methionine analog, L-azidohomoalaine (AHA), for 30 min at 37 °C with shaking, at final concentration of 50 µM. Cells were fixed with 3.7% formaldehyde and permeabilized with 0.1% Triton X-100. Fluorescent labelling was performed for 30 min with Alexa Fluor® 488 reaction cocktail. In both OPP and AHA experiments cells were stained with DAPI.

## Microscopy data analysis

Images were analysed using ImageJ[65] and the statistical analyses were performed in R[66]. Sporulation efficiency was calculated as the ratio between cells with asymmetric septum and cells without, for both WT and 3KO strains ($n > 600$), at 2, 3, and 4 hours post sporulation induction (Supplementary Table 1a). Differences in sporulation efficiency between the strains were statistically examined using Fisher's exact test. For fluorescence localization measurements, a 1p line was drawn across the cells (region of interest, ROI), in the middle. The cell lengths were normalised to 100% and the mean fluorescence intensities across the line were recorded and visualised, error bars represent standard deviation (SD). For WT-RpsB-GFP observations, the following number of cells were measured: T0 – 109, T1 – 110, T2 – 110, T3 – 119, T4 – 116, T5 – 120, T6 – 120 (Fig. 3b). In the OPP assay, the following number of WT cells were measured: T0 – 226, T1 – 226, T2 – 226, T3 – 226, T4 – 226, T5 – 208, T6 – 226 (Fig. 3d). For fluorescence intensity measurements, a 6p line was drawn across each cell, in the middle (ROI, $n > 250$). Fluorescence was recorded and mean values were calculated for each cell. This data was used to construct histograms and box-and-whiskers plots. In the OPP assay, the following number of WT cells was calculated: T0 – 263, T1 – 255, T2 – 279, T3 – 270, T4 – 270, T5 – 275, T6 – 187; and the following number of 3KO cells: T0 – 265, T1 – 265, T2 – 278, T3 – 270, T4 – 270, T5 – 275, T6 – 220 (Fig. 5c). In the AHA assay the following number of WT cells was calculated: T0 – 146, T1 – 280, T2 – 280, T3 – 280, T4 – 280, T5 – 280, T6 – 196 (Fig. 3f); and the following number of 3KO cells: T0 – 121, T1 – 206, T2 – 260, T3 – 260, T4 – 260, T5 – 260, T6 – 260 (Fig. 5e). To compare the distribution of fluorescence intensities between WT and 3KO strains in the OPP assay, two-sided Kolmogorov–Smirnoff test was used. To investigate differences between the fluorescence intensities in the AHA assay between the consecutive timepoints for each strain, two-sided Kruskal–Wallis test with Dunn's post-hoc tests with Bonferroni correction was used.

## Ribosome purification and mass spectrometry

Ribosomes were purified from WT and 3KO strains harvested at 3 h post sporulation induction. Cells were harvested by filtration and flash freezing and the lysates were prepared as described above. Lysates were separated on 5–30% sucrose gradients prepared in polysome buffer (20 mM Tris-HCl, 10 mM MgAcet, 5 mM KAcet, 100 mM NH$_4$Cl) by ultracentrifugation in a Thermo Scientific Sorval WX Ultra Series ultracentrifuge for three hours at r$_{max}$ of 134.161 × $g$ at 28.000 rpm, 4 °C using a Thermo Scientific Th-641 swinging bucket rotor. Samples were collected using Biocomp fractionator based on UV profile and monosome containing fractions were collected for mass spectrometry analysis. The collected samples were subjected to chloroform/methanol precipitation and the resulting protein pellets were washed with methanol. The pellets were air-dried for 10 min and resuspended in 100 mM HEPES pH 8.0. TCEP (final conc. 10 mM) and chloroacetamide (final conc. 40 mM). Trypsin was added in a 1:100 enzyme-to-protein ratio and incubated overnight at 37°. The digestion was terminated by the addition of trifluoroacetic acid (TFA, final conc. 1%). The resulting peptides were labelled using an on-(StageTip) column TMT labelling protocol[67]. Peptides were eluted with 60 µl 60% acetonitrile/0.1% formic acid in water. Equal volumes of each sample were pooled into two TMT12plex samples and dried using a SpeedVac concentrator. Peptides were then loaded on StageTip columns and washed with 0.1% TEA/5%ACN, eluted with 0.1% TEA/50%ACN, and dried using a SpeedVac concentrator. Prior to LC-MS measurement,

the peptide fractions were dissolved in 0.1% TFA, 2% acetonitrile in water. Chromatographic separation was performed on an Easy-Spray Acclaim PepMap column 50 cm long × 75 μm inner diameter (Thermo Fisher Scientific) at 55 °C by applying 180 min acetonitrile gradients in 0.1% aqueous formic acid at a flow rate of 300 nl/min. An UltiMate 3000 nano-LC system was coupled to a Q Exactive HF-X mass spectrometer via an easy-spray source (all Thermo Fisher Scientific). The Q Exactive HF-X was operated in TMT mode with survey scans acquired at a resolution of 60,000 at m/z 200. Up to 15 of the most abundant isotope patterns with charges 2–5 from the survey scan were selected with an isolation window of 0.7 m/z and fragmented by higher-energy collision dissociation (HCD) with normalized collision energies of 32, while the dynamic exclusion was set to 35 s. The maximum ion injection times for the survey scan and the MS/MS scans (acquired with a resolution of 45,000 at m/z 200) were 50 and 96 ms, respectively. The ion target value for MS was set to 3e6 and for MS/MS to 1e5, and the minimum AGC target was set to 1e3. Data was processed with Max-Quant v. 1.6.17.0[68], and peptides were identified from the MS/MS spectra searched against a modified Uniprot *Bacillus subtilis* reference proteome (UP000001570 was extended by four entries: protein RpsN1 sequence MAKKSMIAKQQRTPKFKVQEYTRCERCGRPHSVIRKFK LCRICFRELAYKGQIPGVKKASW; protein RpsN2 sequence MAKKSKVA KELKRQQLVEQYAGIRRELKEKGDYEALSKLPRDSAPGRLHN RCMVTGRP RAYMRKFKMSRIAFRELAHKGQIPGVKKASW; protein RpmG2 sequence MRKKITLACKTCGNRNYTTMKSSASAAERLEVKKYCSTCNSHTAHLETK; protein RpmG3 sequence MRVNVTLACTETGDRNYITTKNKRTNPDR-LELKKYSPRLKKYTLHRETK) using the built-in Andromeda search engine. Reporter ion MS2-based quantification was applied with reporter mass tolerance = 0.003 Da and min. reporter PIF = 0.75. Normalization was selected, weighted ratio to all TMT channels. Cysteine carbamidomethylation was set as a fixed modification and methionine oxidation as well as glutamine/asparagine deamination were set as variable modifications. For in silico digests of the reference proteome, cleavages of arginine or lysine followed by any amino acid were allowed (trypsin/P), and up to two missed cleavages were allowed. The false discovery rate (FDR) was set to 0.01 for peptides, proteins, and sites. A match between runs was enabled. Other parameters were used as pre-set in the software. Unique and razor peptides were used for quantification enabling protein grouping (razor peptides are the peptides uniquely assigned to protein groups and not to individual proteins). Reporter intensity corrected values for protein groups were loaded into Perseus v. 1.6.10.0[69]. Standard filtering steps were applied to clean up the dataset: reverse (matched to decoy database), only identified by site, and potential contaminant (from a list of commonly occurring contaminants included in MaxQuant) protein groups were removed. Next, all proteins except for the ribosomal proteins were removed. Reporter intensity corrected values were log2 transformed, normalized by median subtraction within the samples (TMT channels), and Log2FC 3KO vs WT values were calculated.

## Reporting summary

Further information on research design is available in the Nature Portfolio Reporting Summary linked to this article.

## Data availability

The mass spectrometry proteomics data generated in this study have been deposited in the ProteomeXchange Consortium[70] via the PRIDE[71] partner repository under accession code PXD047497. The RNA-seq and RIBO-seq data generated in this study have been deposited in the NCBI's Gene Expression Omnibus[72] database under accession code GSE249450. Source data are provided with this paper.

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

## Acknowledgements

A.L.S. would like to acknowledge the financial support of EMBO (Installation Grant 3914), and POIR.04.04.00-00-3E9C/17-00 carried out within the First TEAM programme of the Foundation for Polish Science co-financed by the European Union under the European Regional Development Fund.

## Author contributions

O.I. performed experiments and data analysis and wrote the manuscript; P.L. performed data analysis and prepared figures; O.I., N.K. and M.K. prepared RNA-seq and RIBO-seq libraries and material for microscopic observations; N.K. performed growth and germination assays; M.L. performed confocal microscopy; R.S. performed mass spectrometry; A.L.S. conceived and supervised the project.

## Competing interests

The authors declare no competing interests.
