## [Peer Review File · Nature Communications]

Translation in *Bacillus subtilis* is spatially and temporally coordinated during sporulationReviewer #1 (Remarks to the Author):

This manuscript analyses the temporal and spatial pattern of translation during the process of sporulation in *B. subtilis*. Employing a spectrum of approaches including ribosome profiling and fluorescence microscopy, the authors provide a comprehensive set of data that will be of interest to investigators in the field. My overall enthusiasm is lessened, however, by a number of substantial issues (see below) including questions about the proper interpretation of the data as well as incorrect citations to the literature. In addition, the paper fails to make a sufficient argument that translation is an important point of regulation in sporulation, that is, is it more than subsidiary to the (very well-studied) mechanisms of transcriptional regulation? This is particularly important given the tight correlation between transcription and translation reported here (Fig 1, line 105).

1. They state that “...the lower overall TE in T6 and T7 suggest that the cell progressively silences translation during sporulation and by the end, the ribosomes are paused at the SD sequences” (line 177). There are two issues. First, these differences (Fig. 2A) do not seem to be that substantial, and do not appear to be consistent with the data they present in Fig. 3. A possible explanation of this lack of effect is that the TE measurements take into account (as I understand it) genes in both the mother cell and the forespore – presumably the mother cell is not experiencing a similar shutdown. So, redoing their analysis focusing on the differential TE of forespore and mother cell genes might be illuminating. Second, this is an overinterpretation – while it is true that translation elongation appears to be reduced (not surprisingly) this data does not show “at the end” by which a reader reasonably assumes to mean the end of sporulation – that is a dormant spore. If they want to claim that ribosomes remain attached to SD sequences during dormancy (and this would be very interesting), such a claim requires more evidence than is provided.

2. A potential complication in interpreting the data in Fig. 2B is that T0 and T1 contain many of the most highly translated mRNAs in the cell (e.g., those encoding ribosomal proteins) that are absent from the later time points, so there are qualitative differences in the mRNA pool that may account for the overall differences in the shapes of the ribosome density distributions

3. The fluorescence experiments in Fig. 3 & 5 are difficult to interpret in the light of possible changes to permeability to either OPP (puromycin) or AHA (methionine). As the authors themselves note, changes in the spore coat may affect permeabilities (line 226) and “cells experiencing starvation due to sporulation induction resulting in an increased AHA transport into the cell (line 242)”. (Incidentally, there is no reference given for this point, which seems quite relevant, if true). These changes are key to interpreting various observations, e.g., there is limited signal in forespore at T4/5 (Fig. 3C), but as the authors note, the forespore is not translationally inactive at this point (e.g., “Very high expression values of the late sporulation genes in cluster 8 can be assigned to the spore σ G regulon, which points to a high translational activity of the spore” (line 422.) Also, labeling the forespore post-engulfment is likely challenging as the compounds need to cross three membranes (cytoplasmic, outer forespore, inner forespore). This may explain the lack of signal seen in Fig. 3C, T3-T7, and finally, it is not clear why the two probes should provide rather different patterns of labeling; they both should be labeling nascent chains.

4. The experiments in Fig. 4 examining the sporulation delay of the 3KO strain (lacking the three ribosomal paralogs) indicate a global delay in sporulation initiation, as the authors note (line 312). If this is the case, then all their analysis of later time points (>T0) is not likely to reveal anything specific about the mechanism underlying the mutant phenotype. That, any later

effects are probably a consequence of translational mis-regulation early. The authors need to look at compartment-specific reporters of gene expression (as was done in Feaga et al., 2023) which revealed that in the case of an EF-P mutant, the delay was early.

Line 42: Sporulation is not irreversible (at least until sigmaF activation, see Dworkin & Losick 2005). Commitment is not at asymmetric septum formation (line 48).

Line 64: Ref #15 is about quiescence; it deals with sporulation and should be discussed separately.

Line 75: Feaga, et al., 2023 show that EF-P is important, not “necessary.”

Line 404: Ref #16 is not correct in this context

Line 426: Some of these references refer to germination not sporulation, and also none suggest (to my knowledge) any transcriptional activity in the dormant spore.

Reviewer #2 (Remarks to the Author):

This manuscript uses a deletion mutant of three zinc-independent ribosomal proteins, L31*, L33*, and S14*, in *Bacillus subtilis*, to probe their role in sporulation. The authors use RNA-seq, ribosome profiling, and microscopy to show that the three ribosomal proteins may be involved in regulating metabolic and sporulation pathways, translational silencing, and germination. Although the topic of zinc-independent ribosomes and their role in sporulation are important, the manuscript suffers from several fatal flaws, as described below.

Transcriptional analysis should be validated (i.e., by RT-PCR or WB), especially since there were only two samples per condition. Therefore, validation of several main genes/pathways is needed. The number of samples and the number of independent experiments should be noted for all results presented in the paper. Appropriate statistical analysis should be applied (e.g., S Table 1).

As the authors noted, rpmGC is a pseudogene, so what is the rationale behind deleting this gene?

Lines 283-288 – The purpose of the MS analysis and the interpretation of the results are not clear. Was the WT in Supp. Data 2 indeed the WT strain or WT construct with mcherry-tagged L31*? In the way it was written, it seemed like the mcherry construct was verified by MS, but it is not clear how. L31* could be free (i.e. not bound to ribosomes), especially if it is tagged. Were purified ribosomes fluorescent?

(“rpmE2” in Sup. Data 2 is presumably L31* or “RpmEB”, as written in Supp. Fig 3. It would be helpful to keep consistent naming and use the proper convention for writing protein and gene names.)

If zinc-independent ribosomal proteins are at “low expression levels” in the ribosomes (line 284), how can be any significant differences between the WT and the KO mutant due to presence/absence of these proteins? (Note again that L33* will not be expressed anyway because it is a pseudogene. This reviewer also disagrees that paralogs are too similar to be

distinguished by MS.) There is no evidence that zinc-independent RPs are sufficiently expressed and the authors state that zinc is likely not limited (Line 471), which is the main driver for expression of zinc-independent paralogs (the other factors being poorly defined and/or controversial). To be fair, the authors say that zur regulon is not upregulated in the KO mutant and they do not mention anything about zur in the WT. However, there is no reason to think that WT is different regarding zinc limitation if grown under the same conditions as the KO mutant. In fact, the mutant should demand more zinc (as it has to build ribosomes with zinc-dependent paralogs) and it would be more zinc limited, thus zur regulon should be more upregulated. Considering that the authors justify studying these ribosomal proteins because they are expressed under zinc limitation, it is not clear if the conditions they used actually resulted in the sufficient expression of the proteins in question in order to compare WT to the KO mutant. In addition, the authors did not include a complementing strain, so it is possible that any differences observed in this study are due to off-target effects of the deletions. In summary, without proper controls and validations, this study is not sufficiently rigorous to provide reliable data and conclusions. Even if the methodology is improved, the study is mostly descriptive without mechanistic insights.

Translation in Bacillus subtilis is spatially and temporally coordinated during sporulation

Olga Iwańska^{1,+}, Przemysław Latoch^{1,+}, Natalia Kopik², Mariia Kovalenko¹, Małgorzata Lichočka¹, Remigiusz Serwa³, Agata L Starosta^{1,*}

Response to the reviewer's comments:

Reviewer #1

This manuscript analyses the temporal and spatial pattern of translation during the process of sporulation in *B. subtilis*. Employing a spectrum of approaches including ribosome profiling and fluorescence microscopy, the authors provide a comprehensive set of data that will be of interest to investigators in the field. My overall enthusiasm is lessened, however, by a number of substantial issues (see below) including questions about the proper interpretation of the data as well as incorrect citations to the literature. In addition, the paper fails to make a sufficient argument that translation is an important point of regulation in sporulation, that is, is it more than subsidiary to the (very well-studied) mechanisms of transcriptional regulation? This is particularly important given the tight correlation between transcription and translation reported here (Fig 1, line 105).

1. They state that "...the lower overall TE in T6 and T7 suggest that the cell progressively silences translation during sporulation and by the end, the ribosomes are paused at the SD sequences" (line 177). There are two issues. First, these differences (Fig. 2A) do not seem to be that substantial, and do not appear to be consistent with the data they present in Fig. 3. A possible explanation of this lack of effect is that the TE measurements take into account (as I understand it) genes in both the mother cell and the forespore – presumably the mother cell is not experiencing a similar shutdown. So, redoing their analysis focusing on the differential TE of forespore and mother cell genes might be illuminating.

Fig. 2a shows how the distributions of TE of all the genes shift in time, from the logarithmic growth to late sporulation. Our aim here was to show overall changes in translation efficiency that the cell undergoes during the process of sporulation. In this figure we did not intend to focus on specific sigma regulons, or specific genes, as this was to some extent shown in Fig.1. Moreover, Fig. 2a shows a clear trend in the TE data (distributions move to the left and broaden), which we do not find insubstantial. We consider the existing tools for TE analysis somewhat lacking in terms of global trends/differences analysis which is why we decided not to proceed in that direction. On the other hand, formal statistical analysis has been used in the literature, e.g. in Yang *et al.* (2021), who performed Wilcoxon test to compare the overall TE between their samples. Due to specificity of the RNA-seq and RIBO-seq data, we are hesitant towards the use of "classical statistics" for TE analyses. However, we did perform Kruskal-Wallis test (data not normally distributed, $p < 0.05$ in Shapiro-Wilk test) which was statistically significant ($p < 0.05$) and the post-hoc tests (Dunn's test with Bonferroni correction) revealed that the differences in overall TE between the timepoints were statistically significant for all pairs, except for WT6-WT7, WT4-WT7 and WT4-WT6. Therefore we would like to keep both Fig.2a and the statement that TE decreases over time (lines 167, 168).

Regarding the differential TE analysis for mother cell and forespore genes – as mentioned above, we have already done a similar type of analysis and this data is presented in Figure 1c, where we look at the correlation of fold changes in mRNA and ribosomal footprints for genes in two pairs of sigma regulons – mother cell's E and K, and forespore's F and G – between consecutive timepoints. We also looked at the genes exhibiting high ribosome density in their 5' UTR regions in late sporulation (T5-T7) and showed that these belong to late sporulation sigma regulons G and K (forespore and mother cell) (line 174).

We are unclear regarding the Reviewer's comment on data consistency between Fig.2a and Fig.3. Figure 3 shows levels and localisation of translation during sporulation in *B. subtilis* using fluorescence

microscopy and click-chemistry based assays. We show data for active translation (observed as termination of translation by puromycin analogue and labelling of the nascent polypeptide chain using OPP-based assay; this allows for observation of translation in the location of the polypeptide chain synthesis) and new peptide synthesis (observed as a continuous incorporation of methionine analogue AHA; this assay allows polypeptide chains to fold into the nascent proteins and then to be transported/diffused to the designated location in the cell). In both assays we detect decreased levels of fluorescence in later time points (T4-T6), suggesting decreased incorporation of tRNA and decreased amount of newly synthesised peptides. This in fact stays in agreement with the decreasing translation efficiency data shown in Figure 2a – overall TE values decrease gradually from T0 (logarithmic growth) to T7 (seven hours post sporulation induction).

2. Second, this is an overinterpretation – while it is true that translation elongation appears to be reduced (not surprisingly) this data does not show “at the end” by which a reader reasonably assumes to mean the end of sporulation – that is a dormant spore. If they want to claim that ribosomes remain attached to SD sequences during dormancy (and this would be very interesting), such a claim requires more evidence than is provided.

We acknowledge that the language used in line 177 was unclear. We have now changed this to the following (line 178):

- “**Very high ribosome coverage at the SD sequences**, together with the lower overall TE in T6 and T7 suggest that the cell progressively silences translation **in late sporulation.**”

We do not claim that the ribosomes remain attached to the SD sequences during dormancy, as this lies outside of the scope of this work. We state that during sporulation, translation gradually decreases: translation elongation is indeed reduced and translation initiation is prolonged, as suggested by enrichment of footprints at the SD sequences.

3. A potential complication in interpreting the data in Fig, 2B is that T0 and T1 contain many of the most highly translated mRNAs in the cell (e.g., those encoding ribosomal proteins) that are absent from the later time points, so there are qualitative differences in the mRNA pool that may account for the overall differences in the shapes of the ribosome density distributions

The aim of Fig.2 was to show global changes in translation efficiency during the process of sporulation in *B. subtilis*. Panel b provides more detail regarding global translation, as we are primarily focusing on the positions of the ribosomes on the mRNA transcript and not the amount of mRNA or the amount of footprints. In fact, genes with the highest expression in late sporulation (belonging to cluster 8) do not exhibit uniform footprint distribution. However, we acknowledge that this may not be clear and we have now provided additional Supplementary Fig. 3 showing ribosomal footprint distribution for each cluster (Cluster 1-8 from Figure 1a). Since clusters are segregated based on similar expression patterns, and we also showed them to be segregated based on functionality, the additional Supplementary Fig. 3 shows ribosome footprint distribution for early and late sporulation related genes in both forespore and mother cell, in each timepoint. Figure 2b shows distribution plots for each timepoint, for all genes and below we show the footprint distribution plots for all timepoints, but without the genes encoding ribosomal proteins.

Fig.R1. Mean ribosome density on CDSs with 5' and 3' UTRs (+/- 50 nt) during sporulation in *B. subtilis* normalized to the maximum peak value, without 60 genes encoding ribosomal proteins, based on SubtiWiki annotation.

In Figure 2b the RIBO-seq data was normalised using standard normalisation methods. Next, each gene was divided into the same number of bins (gene length normalisation) and footprint densities were also normalised for each timepoint (maximum peak normalisation). We have now also clarified the materials and methods section as follows (line 597 after correction): “Furthermore, average coverage plots for all genes and clusters of genes were produced using the DeepTools⁶¹ library using BPM normalization (bins per million mapped reads).”

4. The fluorescence experiments in Fig. 3 & 5 are difficult to interpret in the light of possible changes to permeability to either OPP (puromycin) or AHA (methionine). As the authors themselves note, changes in the spore coat may affect permeabilities (line 226) and “cells experiencing starvation due to sporulation induction resulting in an increased AHA transport into the cell (line 242)”. (Incidentally, there is no reference given for this point, which seems quite relevant, if true).

We do agree with Reviewer’s #1 comment regarding challenges in the interpretation of the Click-chemistry based experiments, especially in the context of spore permeability in late sporulation. We included this in the original submission:

- “Localisation of the newly synthesised proteins shifts from inside of the spore to the spore coat, which may at least partly result from an impervious spore coat and poor fluorescent dye internalisation. “ (lines 261-263 after correction)
- “It should be noted however, that due to impermeability of the mature spore coat, the internalisation of the fluorescent dye may be inadequate in late sporulation, and the collected data might represent translation in the mother cell more adequately than in the spore during late sporulation (T6)” (lines 460 – 463 after correction).

We believe that as long as we are aware of and upfront with the limitations of the methodology used, and consider them in data interpretation, the results shown here are meaningful and valid.

In terms of new protein synthesis one hour post sporulation induction and transport of AHA into the cells (line 242) – in this part of the manuscript we are describing the results obtained in the click-chemistry based experiment and also discussing possible implications. We are not simply saying that the increase in both fluorescence and range are directly related to changes in protein synthesis, but we are

hypothesising how the experimental setup could potentially influence the outcome. However, we do acknowledge that the clarity of the message can be improved and we have now changed this fragment as follows (lines 245-251 after correction): “This is most probably a result of proteome adaptation to the conditions of nutrient limitation involving activation of several different nutrient sequestration pathways before the cell commits to sporulation³³. The increased intensities may perhaps also be attributed, in a small degree, to the cells experiencing starvation resulting in an increased AHA transport into the cell. In this transition phase (T1-T2), a smaller subpopulation of cells has an increased fluorescence signal, while the rest slowly suppresses protein synthesis in preparation for sporulation.”

5. These changes are key to interpreting various observations, e.g., there is limited signal in forespore at T4/5 (Fig. 3C), but as the authors note, the forespore is not translationally inactive at this point (e.g., “Very high expression values of the late sporulation genes in cluster 8 can be assigned to the spore σ G regulon, which points to a high translational activity of the spore” (line 422.). Also, labeling the forespore post-engulfment is likely challenging as the compounds need to cross three membranes (cytoplasmic, outer forespore, inner forespore). This may explain the lack of signal seen in Fig. 3C, T3-T7, and finally, it is not clear why the two probes should provide rather different patterns of labeling; they both should be labeling nascent chains.

As can be seen in Figures 3c and 5b (OPP) the fluorescence signal is visible in the spore after the spore engulfment until T4, and in Figures 3e and 5d (AHA), until T5. The microscopic images in Fig. 3 and 5 show cells representative of the entire population and we did additionally present plots of the fluorescence intensities based on measurements of >200 cells per timepoint (Figure 3d and f; Figure 5c and e). We specifically intended to provide both qualitative and quantitative observations of translation localisation and amount of nascent peptides until late sporulation. For this reason, the cells were measured directionally, i.e. each cell was measured from the pole where the asymmetric division/spore was located, to the opposite pole. Based on this quantification data we showed that there is fluorescence signal in T4 inside the spore, and the signal is above baseline (Fig. 3c). We did not include timepoint T7 in these experiments due to technical aspects of the assays, i.e. Alexa labelling in both assays is performed on fixed cells, and in T7 most of the cells are lysed and therefore washed away during the fixing protocol.

As mentioned above, we do agree that in later timepoints (T6, and sometimes T5) the microscopic data may be limited due to the nature of the spore. However, we would like to keep the images in Figures 3 and 5 for completeness of the data points, considering that we did discuss the potential limitations to data interpretation in the discussion section. We also consider the microscopic and sequencing data to be complementary. In our opinion, the fluorescence microscopy data is very interesting as, to our knowledge, the cellular localisation of translating ribosomes and nascent peptides during sporulation using OPP and AHA-based assays has not been published before and is an engaging addition to the field of sporulation in *B. subtilis*.

Regarding the use of two probes – puromycin (OPP) and methionine (AHA) derivatives – we have now included additional explanation regarding the differences between these two assays:

- “As incorporation of AHA does not terminate translation, the AHA-containing nascent chains can fold into proteins and reach their cellular localisations. (lines 232-234 after correction).

As puromycin is an antibiotic that terminates translation by mimicking the 3' end of the aminoacylated tRNA, the ribosome that includes OPP into the nascent chain is halted and translation “freezes” - the ribosome does not finish protein synthesis. For this reason, OPP shows localisation of actively translating ribosomes rather than labelling nascent chains. On the other hand, AHA can be considered as an alternative to radiolabelled amino acids, and in this context AHA indeed labels nascent chains. AHA can be incorporated into the growing peptide by the ribosome, which then continues to synthesise the new protein, as it is not blocked by AHA. Due to incubation times of 30 min with both OPP and AHA we can therefore track where and at what levels the proteins are being synthesised (OPP) as well as their amounts and subcellular destination (AHA).

We have also performed additional control to the microscopic observations – staining of the sporulating cells with Alexa dye, without the use of OPP nor AHA to exclude any unspecific labelling, and with DAPI as a control to show fluorescent dye permeability (Fig.R2.). We show that there is no background and the fluorescence signal is specific for both OPP and AHA until T6 when the spore becomes impermeable also to DAPI. In summary, after engulfment the spore is permeable to the fluorescent dyes (T4 and T5) and becomes impermeable upon spore coat maturation (when the spore becomes phase bright).

Fig.R2. Microscopic images of *B. subtilis* WT cells treated with Alexa 488 and DAPI during six hours of sporulation (T0 – T6). Scale bar is 2 μ m. Spores that are phase bright are impermeable and DAPI and Alexa staining is non-specific. Spores that completed engulfment but are not phase bright are permeable to the fluorescent dyes.

6. The experiments in Fig. 4 examining the sporulation delay of the 3KO strain (lacking the three ribosomal paralogs) indicate a global delay in sporulation initiation, as the authors note (line 312). If this is the case, then all their analysis of later time points (>T0) is not likely to reveal anything specific about the mechanism underlying the mutant phenotype. That, any later effects are probably a consequence of translational mis-regulation early. The authors need to look at compartment-specific reporters of gene expression (as was done in Feaga et al., 2023) which revealed that in the case of an EF-P mutant, the delay was early.

We agree with Reviewer #1 that the 3KO strain (lacking the three ribosomal paralogs) shows delayed sporulation initiation, and we also show the process of sporulation is dysregulated in the mutant throughout the entire sporulation. As the three paralogs are expressed at different time points during sporulation, we believe that investigating later time points (after T0) does provide valuable information. We also performed differential expression analysis for all genes, including compartment-specific reporters, and we found that *spolIGA*, *spolIE* (maturation and control of early sporulation sigma factors σ^E and σ^F) and *sigG* are downregulated in 3KO (lines 311-313 after correction).

Line 42: Sporulation is not irreversible (at least until sigmaF activation, see Dworkin & Losick 2005).

Line 42: We have now deleted the word “irreversible”.

Commitment is not at asymmetric septum formation (line 48).

Line 48: We have now deleted the following: “i.e. form the asymmetric septa, they are committed to the sporulation process.”

Line 64: Ref #15 is about quiescence; it deals with sporulation and should be discussed separately.

Line 64 (line 63 after corrections): We have now added “during or at entry into quiescence”. We do not discuss sporulation here, nor the references we used. Both references were used as examples of regulated protein synthesis during or upon entry into quiescence.

Line 75: Feaga, et al., 2023 show that EF-P is important, not “necessary.”

Line 75: We have now changed ‘necessary’ to ‘important’.

Line 404: Ref #16 is not correct in this context

Line 404 (line 416 after corrections): We have now changed the reference. Changed to: Lee, C. Der & Tu, B. P. Metabolic influences on RNA biology and translation. *Crit. Rev. Biochem. Mol. Biol.* 52, 176–184 (2017).

Line 426: Some of these references refer to germination not sporulation, and also none suggest (to my knowledge) any transcriptional activity in the dormant spore.

Line 426 (lines 437-439 after corrections): Yes, the references relate to germination, as the sentence in line 426 refers to any kind of supposed spore inactivity, not only during sporulation. The purpose of this sentence was to give examples of research showing spore activity during the periods when the spore was considered to be inactive: “In fact, more evidence is emerging indicating spore transcriptional and translation activity during periods of supposed inertness^{13,40,42}.”

Sinai *et al.*, 2015 – “Furthermore, we provide evidence that, in contrast to current thinking, protein synthesis occurs during germination and is essential for its execution.” – translation during germination

Riley *et al.*, 2021 – “proteins directly involved in transcription and translation were present at similar levels in the mother cell and the forespore, consistent with active RNA and protein synthesis occurring in both cells throughout sporulation.” – transcription and translation during sporulation. This paper focuses on metabolic differentiation between the mother cell and the forespore. It shows that the forespore is dependent on the mother cell to provide metabolites such as amino acids, however, both transcription and translation take place in the spore.

Zhou *et al.*, 2023 – “Here, we show that the spore arrays the RNA polymerase (RNAP) complex at designated intergenic promoter regions, ensuring timely expression of vital adjacent genes during germination.” - transcription during germination

Reviewer #2

This manuscript uses a deletion mutant of three zinc-independent ribosomal proteins, L31*, L33*, and S14*, in *Bacillus subtilis*, to probe their role in sporulation. The authors use RNA-seq, ribosome profiling, and microscopy to show that the three ribosomal proteins may be involved in regulating metabolic and sporulation pathways, translational silencing, and germination. Although the topic of zinc-independent ribosomes and their role in sporulation are important, the manuscript suffers from several fatal flaws, as described below.

1. Transcriptional analysis should be validated (i.e., by RT-PCR or WB), especially since there were only two samples per condition. Therefore, validation of several main genes/pathways is needed.

The question of validation of the next generation RNA-sequencing results using real time PCR (qPCR) has been discussed in the literature, with the general conclusion that it is not necessary to validate RNA-seq data using qPCR (reviewed in Coenye, 2021). There are several reasons for this, and the two main include:

- Historically, the necessity to validate gene expression data stems from microarray technology and its limitations. Next generation sequencing however improved reproducibility and bias seen in the microarray data and hence, qPCR validation is not necessary.
- According to a number of studies comparing RNA-seq with qPCR data, only a small fraction (~1.8%) of genes show different expression levels in RNA-seq and qPCR, and these are usually short and poorly expressed genes. Therefore, any reliable validation would require checking the expression of a large number of genes in order to identify those that are in fact expressed at different levels. This is very cost-ineffective and denies the purpose of RNA-seq, especially that we are not basing our results on few selected genes (in which case qPCR validation would be an added value), but rather, we look at a global translational landscape of sporulating *B. subtilis*.

We agree that the study would benefit from an increased number of replicates, which is universally true for the majority of studies, however, due to a large number of timepoints and parallel RNA-seq and RIBO-seq, this would be economically challenging. Our results are comparable to the gene expression data available at *SubtiWiki* website (Pedreira *et al.*, 2022) in the form of Expression Browser (it is not possible to perform formal statistical analysis between our dataset and the *SubtiWiki* dataset).

2. The number of samples and the number of independent experiments should be noted for all results presented in the paper.

The number of samples and experiments is described in appropriate supplementary data: Supplementary Data 5 for sequencing data, Supplementary Table 1 for sporulation efficiency data. The sequencing data (including metadata) has also been deposited in NCBI's Gene Expression Omnibus and are accessible through GEO Series accession number GSE249450 (<https://www.ncbi.nlm.nih.gov/geo/query/acc.cgi?acc=GSE249450>). We have now also added the information about the number of samples and experiments for sequencing in the results section (line 95 after correction). We have also included the exact number of measured cells for each experiment and this is now listed in lines 640 – 643 and 646 – 652.

3. Appropriate statistical analysis should be applied (e.g., S Table 1).

We have now added statistical analysis of the sporulation efficiency of WT vs 3KO using Fisher's test (Supplementary Table 1a).

To investigate differences in the fluorescence intensities in the AHA-Alexa 488 protein synthesis assay between consecutive timepoints for each strain (as shown in Figs. 3f and 5e), we performed a two-sided Kruskal-Wallis test with Dunn's post-hoc tests with Bonferroni correction, which is now noted in lines

237 and 402, as well as in respective Figure legends and Materials and Methods. We also added a new supplementary file (Supplementary Data 2) with summary of these statistical analyses, as well as the summary of Shapiro-Wilk tests to check if data follows normal distribution.

We included exact values of a two-sided Kolmogorov-Smirnoff test (Fig. 5c) in lines 395-396.

4. As the authors noted, *rpmGC* is a pseudogene, so what is the rationale behind deleting this gene?

Indeed, *rpmGC* is a pseudogene in *B. subtilis* WT. However, as shown in the literature (Shin and Helmann, 2016), its expression remains under the control of Zur and it was shown to be induced during zinc depletion in WT. We too see low levels of expression in our sequencing data. Thus, to rule out the possibility of having reads mapped to *rpmGC* without the functional gene product, we decided to delete *rpmGC*.

5. Lines 283-288 – The purpose of the MS analysis and the interpretation of the results are not clear. Was the WT in Supp. Data 2 indeed the WT strain or WT construct with mcherry-tagged L31*? In the way it was written, it seemed like the mcherry construct was verified by MS, but it is not clear how. L31* could be free (i.e. not bound to ribosomes), especially if it is tagged. Were purified ribosomes fluorescent?

Lines 283-288: We agree that a clarification is needed. The WT strain which was used for MS analysis did not have RpmEB-mCherry and thus, the purified ribosomes were not fluorescent. We have now changed this fragment as follows (line 288-291 after correction): “The ribosomal localisation of L31* (untagged) was also verified by mass spectrometry. The fractions containing 70S ribosomes were purified by sucrose density gradient centrifugation from WT and 3KO strains and their protein composition was analysed.”

6. (“rpmE2” in Sup. Data 2 is presumably L31* or “RpmEB”, as written in Supp. Fig 3. It would be helpful to keep consistent naming and use the proper convention for writing protein and gene names.)

“rpmE2” has now been changed to rpmEB in Sup. Data 3. “rpsN1/rpsN2” has now been changed to rpsN/rpsNB in Sup. Data 3. The nomenclature is now consistent.

7. If zinc-independent ribosomal proteins are at “low expression levels” in the ribosomes (line 284), how can be any significant differences between the WT and the KO mutant due to presence/absence of these proteins? (Note again that L33* will not be expressed anyway because it is a pseudogene. This reviewer also disagrees that paralogs are too similar to be distinguished by MS.) There is no evidence that zinc-independent RPs are sufficiently expressed and the authors state that zinc is likely not limited (Line 471), which is the main driver for expression of zinc-independent paralogs (the other factors being poorly defined and/or controversial). To be fair, the authors say that zur regulon is not upregulated in the KO mutant and they do not mention anything about zur in the WT. However, there is no reason to think that WT is different regarding zinc limitation if grown under the same conditions as the KO mutant. In fact, the mutant should demand more zinc (as it has to build ribosomes with zinc-dependent paralogs) and it would be more zinc limited, thus zur regulon should be more upregulated. Considering that the authors justify studying these ribosomal proteins because they are expressed under zinc limitation, it is not clear if the conditions they used actually resulted in the sufficient expression of the proteins in question in order to compare WT to the KO mutant. In addition, the authors did not include a complementing strain, so it is possible that any differences observed in this study are due to off-target effects of the deletions.

We understand the concerns that Reviewer #2 raised regarding zinc depletion conditions and zinc homeostasis in general. The main goal of our study was the investigation of translation during sporulation in *B. subtilis* and we also investigated the zinc-independent paralogs of ribosomal proteins in this context. The rationale behind looking at the zinc-independent paralogs of the ribosomal proteins in this study was not to investigate how the zinc limitation conditions affect ribosome composition but rather, whether the paralogs affect translation and translation regulation in the conditions of nutrient limitation, which in our case is sporulation. We therefore do not discuss Zur-regulated response to zinc limitation in much detail, but rather we mention it to give some context. In this study we aimed to focus on the potential differences in translation that may result from altered ribosome composition. This is a part of the specialised ribosomes hypothesis, according to which a small subpopulation of structurally different ribosomes can have a regulatory role in translation (Xue and Barna, 2012). Therefore, the expression of paralogs does not necessary need to be very high. We acknowledge that zinc depletion conditions are difficult to achieve and were probably not achieved in this study. However, sporulation provides a physiological example of conditions of nutrient limitation and under such conditions the level of expression of the ribosomal paralogs was also physiological, so the question whether the expression was “sufficient” was also part of this study. We show that regulatory genes *spoIIGA*, *spoIIIE* (maturation and control of early sporulation sigma factors σ^E and σ^F) and *sigG* are downregulated in 3KO in early sporulation and also, that translation in the 3KO strain is not silenced to the same degree as in WT, based on the differential gene expression analysis and ribosomal footprint density distribution in late sporulation. This suggests that cells lacking the subpopulation of ribosomes containing zinc-independent paralogs exhibit dysregulated translation in sporulation.

Regarding detection of the RpsN/NB proteins by mass spectrometry, we were able to identify only one short peptide (GQIPGVK, data not shown) that is common for both proteins, despite high abundance of the canonical form. This is most probably due to method limitation which renders identification of the paralog unfeasible in this study.

We agree with Reviewer #2 that complementation experiments are usually unambiguous, however, we consider strain complementation in sporulation as potentially inconclusive due to specificity of the process. Because during sporulation the chromosome becomes highly condensed and only certain genomic regions are transcribed, expression of genes from the typical cloning loci in *B. subtilis* (e.g. *amyE*) may be problematic in sporulation.

In summary, without proper controls and validations, this study is not sufficiently rigorous to provide reliable data and conclusions. Even if the methodology is improved, the study is mostly descriptive without mechanistic insights.

References

1. Coenye T. Do results obtained with RNA-sequencing require independent verification? *Biofilm*. 2021 Jan 13; 3:100043. doi: <https://doi.org/10.1016/j.biofilm.2021.100043>.
2. Pedreira T, Efmann C, Stülke J. The current state of SubtiWiki, the database for the model organism *Bacillus subtilis*. *Nucleic Acids Res*. 2022 Jan 7;50(D1): D875-D882. doi: <https://doi.org/10.1093/nar/gkab943>.
3. Shin, JH., Helmann, J. Molecular logic of the Zur-regulated zinc deprivation response in *Bacillus subtilis*. *Nat Commun* 7, 12612 (2016). <https://doi.org/10.1038/ncomms12612>.
4. Xue, S., Barna, M. Specialized ribosomes: a new frontier in gene regulation and organismal biology. *Nat Rev Mol Cell Biol* 13, 355–369 (2012). <https://doi.org/10.1038/nrm3359>.
5. Yang, X., Song, B., Cui, J., Wang, L., Wang, S., Luo, L., Gao, L., Mo, B., Yu, Y., Liu, L. Comparative ribosome profiling reveals distinct translational landscapes of salt-sensitive and -tolerant rice. *BMC Genomics* 22, 612 (2021). <https://doi.org/10.1186/s12864-021-07922-6>

Reviewer #1 (Remarks to the Author):

The authors have presented a series of responses to my original queries. In some cases, they have appropriately answered; in other cases, as detailed below, less so.

In response to my query #1 regarding apparent inconsistency between Figs. 2 & 3, they state “We are unclear regarding the Reviewer’s comment on data consistency between Fig.2a and Fig.3”, let me clarify. Fig. 2 (A & B) indicate the TE & ribosome occupancy goes down - “gradually decreases” (Line 183) particularly at time points later than T4, but by contrast, Fig 3D shows a dramatic drop in OPP staining at T2. These two patterns are not obviously commensurate.

As far as the issue I raised about permeability (my query #4), the response “We believe that as long as we are aware of and upfront with the limitations of the methodology used, and consider them in data interpretation, the results shown here are meaningful and valid” is not sufficient. My argument that given the limitations it is not possible to have confidence in the interpretation of the observations means that in the absence of appropriate control experiments, such interpretations can’t be made.

With respect to their response to my query #6 regarding the origin of the sporulation delay in the 3KO strain, “As the three paralogs are expressed at different time points during sporulation, we believe that investigating later time points (after T0) does provide valuable information.” I disagree – if the delay begins at $\sim T_0$ (which the authors agree), then the later time points are all indirect effects, and not particularly insightful about the underlying cause. It is not obvious to me what “valuable information” they expect to collect with respect to the later time points given that it is not possible to distinguish direct effects at a particular late time point from indirect effects due to the delay at T0.

Reviewer #2 (Remarks to the Author):

I appreciate that the authors attempted to respond to my critiques, but some issues remain unresolved.

I would normally agree with the authors that RNA-seq and Ribo-seq data do not require validation, if they are sufficient numbers of biological replicates. Are duplicates shown here from two separate experiments, two individual growths prepared at the same time, sampling from the same growth twice for two RNA preps, or any other way they were created? Validation is important not for validating the techniques, but to make sure that results are reproducible between biological replicates and specifying the type of replicate is needed to evaluate the rigor. There can be a significant variation between biological replicates that is not captured by performing the experiment only once (or twice?).

Regarding using fluorescently labeled L31* for ribosome localization – there is still no evidence that the tagged protein incorporates into ribosomes, so how can the authors be sure that they are tracking ribosomes? While MS analysis of purified ribosomes (unsurprisingly*) confirmed incorporation of untagged L31*, this experiment does not help verify that the tagged protein is also incorporated. Fluorescently tagged ribosomal proteins often fail to incorporate into ribosomes and therefore this evidence is critical for using the construct for tracking ribosomes. It is still not clear to me what was the purpose of these two experiments in the first place. As the

authors noted, incorporation of these proteins into ribosomes have been established previously.

On that note, the construction of the mcherry-tagged L31* was not described in the method section.

I am still not sure how can authors claim any phenotype of the 3KO strain to be specific to the deletion when the levels of expression of these genes in WT are low (as they say). If zinc limitation is not achieved, and the expression is triggered by sporulation, evidence should be provided. The difference in expression of sigG and other genes is not evidence of “dysregulation of translation”. After all, the overall difference in transcription between WT and 3KO is modest as a very low cutoff was used (log₂FC of 0.6 – Fig 4). Again, without replicating these experiments, these small differences could be by chance.

Note that if expression of enzymes involved in cell wall and membrane structure and transport are DEGs (318-319), this may have an effect on uptake of AHA and OPP and therefore the differences in fluorescence that is observed between WT and 3KO. Once again, this may not be strong evidence for “imbalanced translation”.

The complementing strain is needed to rule out off-target effect in the mutant. Of course, it could be “problematic”, like the authors state, but even partial complementation would be informative.

Naming of RP genes and proteins is still confusing: S2 protein (line 191, 192, 358, Fig. 3) becomes RpsB protein (Fig. 5, line 447, 535), RpmEB (written as a protein, not rpmEB gene) is alternating with L31*, etc.

Translation in *Bacillus subtilis* is spatially and temporally coordinated during sporulation

Olga Iwańska^{1,+}, Przemysław Latoch^{1,+}, Natalia Kopik², Mariia Kovalenko¹, Małgorzata Lichočka¹, Remigiusz Serwa³, Agata L Starosta^{1,*}

Reviewers' comments:

Reviewer #1 (Remarks to the Author):

The authors have presented a series of responses to my original queries. In some cases, they have appropriately answered; in other cases, as detailed below, less so.

In response to my query #1 regarding apparent inconsistency between Figs. 2 & 3, they state “We are unclear regarding the Reviewer’s comment on data consistency between Fig.2a and Fig.3”, let me clarify. Fig. 2 (A & B) indicate the TE & ribosome occupancy goes down - “gradually decreases” (Line 183) particularly at time points later than T4, but by contrast, Fig 3D shows a dramatic drop in OPP staining at T2. These two patterns are not obviously commensurate.

These two patterns do not commensurate as they do not have to in order to show consistent data. Translation efficiency (TE, measured as a ratio between mRNA and ribosomal footprints), ribosome density distribution and active translation measured in O-propargyl-puromycin based assay do not measure, and therefore do not show, exactly the same thing. They all describe translation, but each in a different way. It is important to understand the differences between these measurements, and their implications.

1. TE only tells us how many ribosomes are present per mRNA transcript. The ribosomes may be stalled, paused, hibernated or actively translating. TE does not inform us about the fact, whether the ribosomes are actively translating or not. Having said that, we did report a decrease in global TE between T0/T1 and T2, as shown below for only these three timepoints for clarity. In fact, the first drop in TE happens at T2, and this is followed by decreasing TE until T7 (Fig. R1.)

Fig. R1. Histograms of translational efficiency of *B. subtilis* during sporulation, calculated at T0 (prior to sporulation induction) to T7 (7h post sporulation induction).

2. Ribosome density distribution tells us where the ribosomes are on a transcript: whether they are stalled at e.g. initiation start site, or evenly distributed on a transcript. It does not tell us if the ribosomes are active or not. Uniform distribution may represent active translation as well as ribosomes hibernating on a transcript. Having said that, we also did show that the ribosome density distribution changes globally and that the ribosomes spend more time at the

translation start sites as the sporulation progresses. Together with the TE values, this indicates that global translation slows down as sporulation progresses (as stated in lines 177 – 181).

Fig. R2. Mean ribosome density on CDSs with 5' and 3' UTRs (+/- 50 nt) during sporulation in *B. subtilis* normalized to the maximum peak value. The red line shows smoothed data using moving averages.

3. O-propargyl-puromycin based assay tells us whether the ribosomes in the cell are accepting aminoacyl-tRNA and producing peptide bonds. Out of the three, this one is the closest proxy to describing active translation in the cell, i.e. the actual catalytic activity of the ribozyme. From all three pieces of evidence (TE, ribosome density distribution and OPP-based biochemical assay) we can deduce that in T2 a smaller number of ribosomes sit on the mRNA (TE), compared to T0 and T1, and more ribosomes begin to spend more time at the translation start sites (ribosome density distribution). The ribosomes appear to be however transiently arrested and are translating at very low levels in T2 (O-propargyl-puromycin based assay). We hypothesize that the rationale for such transient arrest is related to asymmetric division. We propose that during this time several things transpire that are somewhat dependant on each other (listed here in no particular order): asymmetric septation, chromosome translocation, transcription reprogramming (according to genomic loci), loss of polar localisation of the ribosomes and transient translational halt. Regarding sporulation timeline, the translational halt lasts until the two asymmetric compartments are formed and transcription is separated based on dedicated sigma factors. The ribosomes do not dissociate from the mRNA (or into subunits), but are instead paused for a while awaiting the new subcellular localisation. Although at this point we cannot propose a molecular mechanism of such transient translational arrest, this topic is definitely something that we would like to pursue in the near future. We have added the following to the discussion section:

Lines 466 - 469 : **Interestingly, translation silencing at T2 was accompanied by the loss of polar localisation of ribosomes. We hypothesise that such loss of localisation and transient translational halt, together with asymmetric septation and chromosome translocation, are necessary steps during sporulation initiation.**

As far as the issue I raised about permeability (my query #4), the response “We believe that as long as we are aware of and upfront with the limitations of the methodology used, and consider them in data interpretation, the results shown here are meaningful and valid” is not sufficient. My argument that given the limitations it is not possible to have confidence in the interpretation of the observations means that in the absence of appropriate control experiments, such interpretations can’t be made.

The response quoted by Reviewer 1 is fragmentary, out of context and seems to miss a follow up control experiment which we presented in a previous rebuttal.

We did provide an additional control experiment demonstrating permeability of the fluorescent dye (which is in fact a clearly identifiable moment – when the spore becomes phase bright). We also showed that fluorescent labelling in click-chemistry based assays is selective until the spore matures – becomes phase bright. We would like to, once again, call attention to the control experiments below, and emphasise that if the spore presents higher fluorescence than the mother cell (which is in fact the case in T3 and T4, green arrowheads), there is clearly no issue with spore permeability. Moreover and most importantly, we do not interpret data which is ambiguous. Each time the data from later timepoints was discussed in the manuscript, we added a disclaimer stating that the results from later timepoints (after the spore matures) are more representative of the mother cell and may be biased due to permeabilisation issues. As we said before, late timepoints are presented for the reason of dataset completion, they are under no circumstances interpreted definitively and most certainly do not annul the entire experiment.

In order to avoid any possible controversies, we have now provided additional clarification in the manuscript (please see below) as well as added the control experiment in the Supplementary Fig. 4.

Lines 224 – 231: Once the asymmetric division took place, ~~the active translation was~~ resumed and localised mostly to the septum (T3) and then predominantly to the prespore (T4). During late sporulation ~~— when spore engulfment and spore coat assembly occurs —~~ the fluorescent signal decreased, which was expected. **After spore maturation, observed as the spore becoming phase bright, the spore is impermeable to the fluorescent dyes and non-specific labelling of the spore coat may be observed (beginning at T5), rendering data from later timepoints representative of the mother cell rather than the mature (phase bright) spore (Supplementary Fig. 4).**

Lines 265 - 268: At T5 and T6 the mean fluorescence intensity gradually decreases indicating reduced protein synthesis **in the mother cell**. ~~Localisation of the newly synthesised proteins shifts from inside of the spore to the spore coat, which may at least partly result from an impervious spore coat and poor fluorescent dye internalisation.~~

Lines 404 – 406: **3KO** showed higher mean fluorescence which suggests dysregulated translation, specifically disrupted translation silencing in **the mother cell** ~~preparation for dormancy~~.

Fig. R3. Top panel shows microscopic images of *B. subtilis* WT cells treated with OPP and stained with Alexa 488 and DAPI, illustrating active translation during six hours of sporulation (T0 – T6). Scale bar is 2 μ m. Bottom panels show microscopic images of *B. subtilis* WT cells treated with Alexa 488 (but not OPP) and DAPI during six hours of sporulation (T0 – T6). Scale bar is 2 μ m. Spores that are phase bright are impermeable, and DAPI and Alexa staining is non-specific. Spores that completed engulfment but are not phase bright are permeable to the fluorescent dyes.

Green arrowheads show higher Alexa 488 signal in the spore than in the mother cell \rightarrow no problem with spore permeability in later timepoints (T4 and T5). White arrowheads point to engulfed but not mature spore and show lack of non-specific fluorescent labelling. The spore is permeable to fluorescent dye (DAPI inside the spore). Red arrowheads point to mature spore (PHASE BRIGHT) which is impermeable to fluorescent dyes and non-specifically labelled (spore coat labelled with DAPI and Alexa 488).

With respect to their response to my query #6 regarding the origin of the sporulation delay in the 3KO strain, “As the three paralogs are expressed at different time points during sporulation, we believe that investigating later time points (after T0) does provide valuable information.” I disagree – if the delay begins at \sim T0 (which the authors agree), then the later time points are all indirect effects, and not particularly insightful about the underlying cause. It is not obvious to me what “valuable information” they expect to collect with respect to the later time points given that it is not possible to distinguish direct effects at a particular late time point from indirect effects due to the delay at T0.

First of all, there is no delay at T0. Our response as quoted by Reviewer 1 is fragmentary and out of context. At no point did we agree that the delay begins at T0. We specifically said that the mutant strain shows delayed sporulation initiation. T0 corresponds to logarithmic growth, which we stated clearly. We assumed that in the original review, Reviewer 1 made a small, nomenclature error and they really meant sporulation initiation (T1-T2), and we answered accordingly. However, this is clearly not the case and perhaps results from miscomprehension of the experimental setup of sporulation induction – T0 represents logarithmic growth in rich medium, prior to sporulation induction by

resuspension in sporulation medium. At T0 (during logarithmic growth) there is obviously no delay in sporulation and there are no changes in translation as shown by lack of DEGs (Fig. 4a), similar levels of translation as shown by OPP-based assay and similar amounts of newly synthesised proteins (AHA-based assay). Also, we complied to what the reviewer suggested we should have done in terms of data analysis looking into the origin of sporulation delay and showed downregulation of sporulation initiation related genes *spollGA*, *spollE* and *sigG*. In fact, we had presented this data in the original submission.

As the expression of different ribosomal paralogs investigated here happens at different timepoints (after T0) as shown in Fig. R4, it would be perfunctory to disregard it and not to investigate later timepoints. Although we acknowledge the option that the later effects may be indirect, we are still investigating the effects on the process of translation during sporulation and how it is affected in its entirety. We are not simply looking at lack of proteins, but in fact, at a subpopulation of ribosomes that is slightly different and as such may have a role in translation during sporulation. As such, we find "valuable information" in the observation that lack of a subpopulation of ribosomes carrying zinc independent paralogs of ribosomal proteins results in the cell's inability to adequately silence translation in preparation for dormancy which appears to be essential for proper spore formation and germination. We have now added this to discussion section :

Lines 496 – 501: **Therefore, we propose that the lack of the subpopulation of ribosomes carrying zinc independent paralogs of ribosomal proteins results in the cell's inability to adequately silence translation in preparation for dormancy which appears to be essential for proper spore formation and germination.** This may perhaps indicate ribosomal finetuning or a regulatory role of a subpopulation of zinc depleted ribosomes during sporulation and growth arrest in *B. subtilis*.

Fig. R4. Expression profiles of three genes *rpmEB*, *rpsNB*, and *rpmGC* encoding paralogs of ribosomal proteins in WT *B. subtilis* during sporulation. Data shown as mean normalized transcript per million (TPM) values from two RIBO-seq biological replicates for each time point (T0 – T7).

Reviewer #2 (Remarks to the Author):

I appreciate that the authors attempted to respond to my critiques, but some issues remain unresolved.

I would normally agree with the authors that RNA-seq and Ribo-seq data do not require validation, if they are sufficient numbers of biological replicates. Are duplicates shown here from two separate experiments, two individual growths prepared at the same time, sampling from the same growth twice for two RNA preps, or any other way they were created? Validation is important not for validating the techniques, but to make sure that results are reproducible between biological replicates and specifying the type of replicate is needed to evaluate the rigor. There can be a significant variation between biological replicates that is not captured by performing the experiment only once (or twice?).

The experiment was performed twice, as in two biological replicates were used to prepare libraries for sequencing. The question of the origin of biological replicates has not been raised by Reviewer 2 before, it is a new comment. As is standard for biological replicates, RNA-seq and RIBO-seq libraries were prepared from two biologically distinct samples – two separate experiments, prepared at different times. Perhaps a close examination of the PCA plots (Fig. R5., Supplementary Fig. 1c and d and Supplementary Fig. 2c and d) may help dispel doubts regarding data reproducibility – the specificity of time series data allows to visualise and predict patterns using principal component analysis, as shown below. Also, we understand the idea that the results need to be verified, not the method, and qPCR or Western blot still remain poor choices to validate the results for the same reasons as discussed before.

Line 96 – 97: **eight samples T0-T7 in biological duplicates, grown and collected on two different days, RNA-seq and RIBO-seq performed in parallel**

Line 308 – 311: To investigate this, we sequenced transcriptome and translome of the 3KO mutant during sporulation, under the same conditions as for WT – from T0 (exponential growth/sporulation induction) to T7 (7h after sporulation induction), in **biological** duplicates (Supplementary Fig. 1 and 2).

Fig. R5. Principal component analysis plots of duplicate samples of the (a) WT transcriptome (RNA-seq), (b) WT translome (RIBO-seq), (c) 3KO transcriptome (RNA-seq), and (d) 3KO translome (RIBO-

seq) data for sporulating *Bacillus subtilis* at different timepoints: 0 to 7 hours post sporulation induction. Black arrows indicate spatial pattern of the time series data in the PCA plots.

Regarding using fluorescently labeled L31* for ribosome localization – there is still no evidence that the tagged protein incorporates into ribosomes, so how can the authors be sure that they are tracking ribosomes? While MS analysis of purified ribosomes (unsurprisingly*) confirmed incorporation of untagged L31*, this experiment does not help verify that the tagged protein is also incorporated. Fluorescently tagged ribosomal proteins often fail to incorporate into ribosomes and therefore this evidence is critical for using the construct for tracking ribosomes. It is still not clear to me what was the purpose of these two experiments in the first place. As the authors noted, incorporation of these proteins into ribosomes have been established previously.

The sole purpose of these experiments was showing that L31* is a ribosomal protein and that it binds to the ribosomes. These experiments are controls showing that during sporulation, L31* is: a) expressed in *B. subtilis*, and b) binds to the ribosomes. As such, they are not in the main text but rather, these are supplementary data and without these, the manuscript would not suffer. No claims were made based on this data, other than simply stating that L31* has ribosomal localisation during sporulation in *B. subtilis*. Having said that, colocalization experiments are gold standard and one of the most widespread applications in fluorescence microscopy. The observation of tagged L31* was not performed in vacuum, we demonstrated that the investigated protein (tagged L31*) occupies the same subcellular region as a reference protein (tagged S2) as shown by the spatial overlap of the two fluorophores (as shown below). If Reviewer 2 is not convinced by the colocalization experiments, there is always MS data to support our observations. Considering all three pieces of information (colocalisation experiments, MS data and literature data), we are confident in stating that L31* is expressed during sporulation in *B. subtilis* and it binds to the ribosome.

Fig. R6. The localization of RpmEB at 1, 2, 3, 4, and 5 hours post-sporulation induction. The ribosomal proteins were tagged with fluorescent protein tags – RpsB-GFP (green) and RpmEB-mCherry (red) in the WT background. The scale bar is 2 μ m.

The plots show the fluorescence intensity profiles measured along the yellow bars from the RpsB-GFP (green) and RpmEB-mCherry (red) images starting from a higher to a lower point. Peaks of mCherry fluorescence maxima correlate with the GFP fluorescence maxima at T3-T5 and correspond to the ribosomal localisation in the cell during sporulation. At T1 levels of mCherry fluorescence are lower than GFP, corresponding to lower expression of RpmEB during sporulation initiation, which then increases as sporulation progresses.

On that note, the construction of the mCherry-tagged L31* was not described in the method section.

This fragment of strain construction methodology was missing and we have now amended the materials and methods section as well as supplementary materials (Supplementary Table 2b and c).

Lines 558-563: **The WT-RpsB-GFP-RpmEB-mCherry strain with fluorescently tagged RpmEB with mCherry at the C terminus was constructed analogously. The competent WT-RpsB-GFP strain was transformed with a linear construct carrying the mCherry gene and chloramphenicol resistance cassette from the pMCL200 plasmid (Supplementary Table 2b and c). The fusion was performed by a double cross-over in the native locus (*rpmEB*).**

I am still not sure how can authors claim any phenotype of the 3KO strain to be specific to the deletion when the levels of expression of these genes in WT are low (as they say). If zinc limitation is not achieved, and the expression is triggered by sporulation, evidence should be provided.

The evidence of expression of the zinc-independent paralogs of ribosomal proteins during sporulation was provided in the original manuscript, as well as in the rebuttal. To summarise, 1) we showed transcriptome and translome data of the WT strain compared to the deletion strain showing expression of genes throughout the entire process of sporulation, including expression of the paralogs, 2) expression of these genes during sporulation is reported in the literature (e.g. SubtiWiki Expression Browser, please see below), 3) presence of the RpmEB paralog in the monosomal fraction collected from sporulating culture was shown here with MS, 4) additional evidence of increasing expression of RpmEB (L31*) during sporulation can be provided using fluorescence microscopy data. Expression of the fluorescently tagged protein (from the native loci) can be measured and compared between timepoints and although such analysis was not included in the original submission, we would be more than happy to provide one if requested, since the data has already been collected (above).

Also, not all genes exhibit high levels of expression (be it transcription or translation) and we are sceptical regarding the implications of Reviewer's 2 comment, that only genes with high expression values can influence the phenotype. Levels of transcription and/or translation depend on e.g. protein stoichiometry and even a small number of copies of a single protein may be essential and sufficient for the cell. Moreover, as we explained before, when looking at specialised ribosomes we expect only a small subpopulation of ribosomes to have different protein content. Therefore, high expression values of the paralogs were neither expected nor necessary in this case.

Fig. R7. Gene expression levels of *rpmEB*, *rpsNB* and *rpmGC* during sporulation (yellow box S1-S8 – 1 to 8 hours post sporulation induction) of *Bacillus subtilis* according to SubtiWiki Expression Browser.

The difference in expression of sigG and other genes is not evidence of “dysregulation of translation”. After all, the overall difference in transcription between WT and 3KO is modest as a very low cutoff was used (log2FC of 0.6 – Fig 4). Again, without replicating these experiments, these small differences could be by chance.

There are several issues with this comment. First, this is a gross oversimplification of our results and discussion. At no point did we state that differentially expressed “*sigG* and other genes” are evidence of dysregulated translation. We provided several pieces of evidence, applied different methodologies and included statistical analyses to arrive at the conclusions presented in the manuscript. Second, please note that we report here translato~~me~~, not transcriptome. The difference is subtle, but important. Third, biological replicates WERE used in the sequencing experiments. Moreover, the results of sequencing are in line with the microscopic observations. Hence, there is little possibility that the differences we reported here are by chance.

Note that if the expression of enzymes involved in the cell wall and membrane structure and transport are DEGs (318-319), this may have an effect on an uptake of AHA and OPP and therefore the differences in fluorescence that is observed between WT and 3KO. Once again, this may not be strong evidence for “imbalanced translation”.

Once again, this is a new comment. Also, it is inconsistent with the interpretation of sequencing data described by Reviewer 2 above – in a previous paragraph DEGs represented insufficient evidence as the cutoff was very low, while in the current paragraph DEGs are sufficient enough to disregard the entire assay. Also, methionine transporter genes (*metNPQ*) were not differentially expressed between 3KO and WT.

In this manuscript, microscopy-based assays (and not only click-chemistry assays) were coupled with RNA-seq and RIBO-seq experiments and the results were interpreted together. Both pieces of evidence (microscopic observations and sequencing data analysis) show the same pattern which we do not perceive as a coincidence.

The complementing strain is needed to rule out off-target effect in the mutant. Of course, it could be “problematic”, like the authors state, but even partial complementation would be informative.

No, it would not, as the results would be inconclusive. If the complementing strain still shows delayed sporulation and dysregulated translation it does not imply that there is no effect in the KO strain due to several reasons. One of them being expression of particular loci during sporulation in *Bacillus subtilis* as discussed earlier. Complementation may work, but it does not have to due to the specificity of the investigated biological process, and such results cannot be unambiguously interpreted.

Naming of RP genes and proteins is still confusing: S2 protein (line 191, 192, 358, Fig. 3) becomes RpsB protein (Fig. 5, line 447, 535), RpmEB (written as a protein, not rpmEB gene) is alternating with L31*, etc.

This is standard biological nomenclature of ribosomal proteins (S2, L31) and standard nomenclature of *Bacillus subtilis* proteins used in the literature. We did change the nonstandard naming of the paralogs (RpmE2, RpsN1/N2). However, we have now unified the nomenclature and use *Bacillus subtilis* protein names.

Reviewer #1 (Remarks to the Author):

The response to my critique is sufficient.

Reviewer #2 (Remarks to the Author):

This is the third review of the same article and I do not have any additional concerns.